# Non-ionotropic NMDAR signalling activates Panx1 to induce P2X4R-dependent long-term depression in the hippocampus

Allison C. Nielsen, Connor L. Anderson, Carina Ens, Andrew K. J. Boyce and Roger J. Thompson 

*Department of Cell Biology & Anatomy, Hotchkiss Brain Institute, Cumming School of Medicine, University of Calgary, Calgary, AB, Canada*

Handling Editors: Peying Fong & Jorge Contreras

The peer review history is available in the Supporting Information section of this article (https://doi.org/10.1113/JP285193#support-information-section).

**Abstract figure legend** Stimulation of the Shaffer collateral pathway at 3 Hz activates non-ionotropic NMDA receptors. This subsequently leads to Src kinase mediated phosphorylation of pannexin-1 (Panx1), which releases ATP from the postsynaptic cell. ATP acts as a ligand for P2X4 receptors to induce long-term depression (LTD).

**Allison Nielsen** has a PhD from University of Calgary with a specialty in neuroscience. Her thesis work focussed on synaptic plasticity and the role of non-ionotropic NMDA receptors and pannexin-1 channels.

**Abstract**   In recent years, evidence supporting non-ionotropic signalling by the NMDA receptor (niNMDAR) has emerged, including roles in long-term depression (LTD). Here, we investigated whether niNMDAR-pannexin-1 (Panx1) contributes to LTD at the CA3–CA1 hippocampal synapse. Using whole-cell, patch clamp electrophysiology in rat hippocampal slices, we show that a low-frequency stimulation (3 Hz) of the Schaffer collaterals produces LTD that is blocked by continuous but not transient application of the NMDAR competitive antagonist, MK-801. After transient MK-801, LTD involved pannexin-1 and sarcoma (Src) kinase. We show that pannexin-1 is not permeable to $Ca^{2+}$, but probably releases ATP to induce LTD via P2X4 purinergic receptors because LTD after transient MK-801 application was prevented by 5-BDBD. Thus, we conclude that niNMDAR activation of Panx1 can link glutamatergic and purinergic pathways to produce LTD following low frequency synaptic stimulation when NMDARs are transiently inhibited.

(Received 1 April 2024; accepted after revision 28 November 2024; first published online 23 December 2024)

**Corresponding author** R. J. Thompson: Department of Cell Biology & Anatomy, Hotchkiss Brain Institute, Cumming School of Medicine, University of Calgary, Calgary, AB T2N 4N1, Canada.    Email: rj.thompson@ucalgary.ca

**Key points**

- Differential effect of short-term D-APV and MK-801 application on long-term depression (LTD) suggests that the NMDA receptor (niNMDAR) contributes to later phases of synaptic depression.
- niNMDAR LTD involved sarcoma (Src) kinase and pannexin-1 (Panx1), which is a pathway previously identified to be active during excitotoxicity.
- Panx1 was not calcium permeable but may contribute to late phase LTD via ATP release. Panx1 blockers prevent LTD, and this was rescued with exogenous ATP application. Inhibition of LTD with 5-BDBD suggests the downstream involvement of postsynaptic P2X4 receptors.

## Introduction

Neuroplasticity is an umbrella term used to describe the brain's ability to adapt and change throughout life in response to experience, stimuli, and environmental conditions. These changes can happen on a large, global network scale (Chowdhury & Hell, 2018; Ho et al., 2011) down to individual synapses (Citri & Malenka, 2008; Mateos-Aparicio & Rodríguez-Moreno, 2019). Activity-dependent changes at the synaptic level are termed synaptic plasticity. Synapses can either increase (potentiation) or decrease (depression) in strength. Changes in synaptic strength that persist for hours to days are considered long-term (Santini et al., 2014) and are critical to physiological processes such as learning and memory (Abraham et al., 2019; Citri & Malenka, 2008; Goda & Stevens, 1996; Malenka & Bear, 2004; Pastalkova et al., 2006; Santini et al., 2014; Tang et al., 1999; Whitlock et al., 2006). The best-characterized forms of long-term plasticity are hippocampal NMDA receptor (NMDAR)-dependent long-term potentiation (LTP) and long-term depression (LTD) (Abraham et al., 2019; Citri & Malenka, 2008; Goda & Stevens, 1996). LTP and LTD are classically modelled as differential rises in calcium ($Ca^{2+}$) influx directly through the NMDAR (Chater & Goda, 2014; Malenka, 1994). Modest elevation

of intracellular $Ca^{2+}$ preferentially activates a protein phosphatase cascade to produce LTD, whereas LTP results from more substantially elevated intracellular $Ca^{2+}$ and the activation of a protein kinase cascade (Malenka, 1994). Depending on the intracellular $Ca^{2+}$ signal and pathway activated, key downstream proteins (e.g. sites along AMPAR subunits and auxiliary proteins) are either phosphorylated or dephosphorylated (Barria et al., 1997; Malenka, 1994; Mammen et al., 1997) to alter AMPAR kinetics (Banke et al., 2001; Derkach et al., 1999) and presence (Beattie et al., 2000; Carroll et al., 2001) at the synapse (Chater & Goda, 2014).

Key evidence from Nabavi et al. (2013) using distinct NMDAR antagonists has questioned this classical model for the generation of LTD. It was found that LTD was still induced in rat hippocampal slices with 1 Hz low-frequency stimulation (LFS) in the absence of ion flux through the NMDAR. The conclusion was that hippocampal LTD can occur independently of ion flux through the NMDAR. Binding of glutamate to the GluN2 subunit produces a conformational change that activates intracellular signalling pathways (Dore et al., 2015). This type of signalling by the NMDAR was initially called 'metabotropic signalling', but the field has subsequently changed to refer to it as non-ionotropic signalling (niNMDAR). Although the concept of

niNMDAR LTD was challenged (Babiec et al., 2014), several groups have provided support for niNMDAR pathways in synaptic plasticity throughout the brain (Carter & Jahr, 2016; Grey et al., 2016; Stein et al., 2015, 2020) and dendritic spine morphological plasticity (Stein et al., 2015).

In addition to the role of niNMDARs in LTD, our work has shown that niNMDAR pathways contribute to excitotoxicity (Thompson et al., 2008; Weilinger et al., 2012, 2016) through activation of sarcoma (Src) kinase and phosphorylation of the large pore channel, pannexin-1 (Panx1), ultimately leading to cell death. As a result of findings linking Panx1 to LTD (Ardiles et al., 2014; Gajardo et al., 2018), we considered whether niNMDAR-Panx1 could contribute to LTD. Here, using whole-cell recordings, we show that NMDARs play a dual role in LTD. In the early phases, ionotropic NMDARs contribute to depression of evoked synaptic responses, but niNMDARs becomes prominent in later phases. The late phase LTD probably involves release of ATP by Panx1, leading to activation of P2X4 receptors and synaptic depression.

## Methods

### Animals

All animal protocols were approved by the University of Calgary's Animal Care and Use Committee (protocol number AC20-0133) in accordance with the Canadian Council on Animal Care. Hippocampal slices were prepared from male Sprague–Dawley rats (Charles River, Wilmington, MA, USA) between postnatal days 30–40. Rats were housed under a 12:12 h light/dark photocycle with *ad libitum* access to food (Rodent Laboratory Chow; Purina, St Louis, MO, USA) and water.

### Acute hippocampal slice preparation

Rats were anaesthetized by isoflurane inhalation and decapitated. Brains were extracted, blocked and mounted on a vibratome (VT1200S; Leica, Wetzlar, Germany) when submerged in ice-cold slicing solution (in mM): 75 sucrose, 87 NaCl, 25 $NaHCO_3$, 2.5 KCl, 25 glucose, 1.25 $NaH_2PO_4$, 7 $MgCl_2$ and 0.5 $CaCl_2$ saturated with 95% $O_2$/5% $CO_2$. Transverse hippocampal slices were cut at 370 µM and transferred to a holding chamber containing warmed (33–35°C) ACSF (in mM): 120 NaCl, 26 $NaHCO_3$, 3 KCl, 10 glucose, 1.25 $NaH_2PO_4$, 1.3 $MgSO_4$ and 2 $CaCl_2$ constantly bubbled with 95% $O_2$/5% $CO_2$.

### Plasmids

Plasmid encoding wild-type rat Panx1 was cloned into a pRK5 expression vector (pRK5-rPanx1). Myc-tagged rat Panx1 (rPanx1-myc) has been described previously (Bruzzone et al., 2005) and was a gift from Dr C. Naus (University of British Columbia, Vancouver, BC, Canada). Plasmids encoding ERT2-PhoCl-Src were based on the previously described PhoCl construct (Zhang et al., 2017) and were a gift from Dr R. Campbell (University of Alberta, Edmonton, AB, Canada/University of Tokyo, Tokyo, Japan). Enhanced green fluorescent protein plasmid (pEGFP-C1) was commercially purchased (Addgene, Watertown, MA, USA).

### HEK293T transfections

HEK293T cells (ATCC, Manassas, VA, USA) were maintained in Dulbecco's modified Eagle's medium, high glucose, supplemented with 10% foetal bovine serum, 1% penicillin and 1% streptomycin, and housed in a humidified cell culture incubator with 95% $O_2$ and 5% $CO_2$ at 37°C. Cells were seeded in six-well plates until proper adherence and desired confluency (70–80%) was reached. Cells were co-transfected with distinct plasmids containing eGFP and wild-type rPanx1 channels (1:5 ratio), or mixed ratios of rat Panx1-myc (rPanx1-myc) ± PhoCl-Src, where indicated. Transfections were carried out using Lipofectamine 2000 transfection reagent (Thermo Fisher Scientific, Waltham, MA, USA) in accordance with the manufacturer's instructions. Transfections progressed for 24–48 h before cells were split and seeded onto poly-D-lysine (Sigma-Aldrich, St Louis, MO, USA) coated coverslips for a minimum of 1 h prior to experimentation. GFP was used to positively identify transfected cells.

### Electrophysiology

All electrophysiological recordings utilized borosilicate glass microelectrodes (Sutter Instrument Company, Novato, CA, USA) pulled to a tip resistance of 2.5–5 MΩ with a P-1000 Flaming/Brown Micropipette Puller (Sutter Instrument Company). For LTD recordings, hippocampal slices were transferred to a recording chamber following recovery (minimum 1 h) where they were constantly perfused with warmed (30–33°C via an inline heater; Harvard Apparatus, Holliston, MA, USA), oxygenated (95% $O_2$/5% $CO_2$) ACSF at a rate of 1–2 mL $min^{-1}$. Microelectrodes were filled with an intracellular recording solution ('internal solution') containing (in mM): 108 potassium gluconate (KGluc), eight sodium gluconate (NaGluc), 8 KCl, 2 $MgCl_2$, 2.5 $K_2$-EGTA, 10 HEPES, 4 $K_2$-ATP and 0.3 $Na_3$-GTP with a pH of 7.25 (KOH). Some experiments included the SFK inhibitor PP2 (1 µM) or its inactive analogue PP3 (1 µM) in the internal solution. For all other experiments, drugs were included in the extracellular ACSF perfusion solution. CA1 pyramidal cells were voltage-clamped at

−60 mV. Cell viability/stability and access resistance ($R_a$) were monitored for 2–5 min without stimulation. This was followed with 5–15 min of baseline paired-pulse stimulation (PPS) to the Schaffer collaterals (1 ms pulses, 50 ms apart/20 s sweep). LFS consisted of a 5-min (900 pulses) 3 Hz stimulation (three evenly spaced 1 ms pulses/1 s sweep). To promote NMDAR activity, cells were held at a slightly depolarized potential (−40 mV) during this time. Following LFS, cells were subjected to a prolonged PPS (30–60 min) where the evoked EPSC peaks were monitored and compared to baseline values. Extracellular drugs were perfused after 5 min of baseline PPS and washed out following LFS. $R_a$ was monitored throughout the entire recording and never reached 25 MΩ. Changes in $R_a$ were kept below 20% of break-in value. Cells were discarded if they failed to meet either of these criteria.

For ion replacement studies, positively transfected HEK293T cells were whole-cell, patch clamped and held at −60 mV. Cells were perfused with extracellular solution at a rate of 1–2 mL min$^{-1}$ and maintained at 30–33°C (inline heater; Harvard Apparatus). Extracellular solution was composed of (in mM): 140 NaCl, 3 KCl, 2 MgCl$_2$, 2 CaCl$_2$, 10 Hepes and 10 glucose with a pH of 7.4 (NaOH). During Ca$^{2+}$ permeability experiments, recordings were performed in 2 mM extracellular Ca$^{2+}$ prior to switching to a high 40 mM extracellular Ca$^{2+}$ solution. The intracellular solution for HEK293T cells contained (in mM): 120 CsCl, 1 EGTA, 1 Hepes, 2 MgCl$_2$, 2 Mg-ATP and 0.3 Na$_3$-GTP at pH 7.3. Cells were voltage-ramped every 1.5 s under both normal (2 mM) and high (40 mM) extracellular Ca$^{2+}$. Current–voltage (*I–V*) curves were generated, and the reversal potential ($E_{rev}$) was determined as the voltage where no current was measured.

### Engineering of a photoactivatable Src kinase

To test whether Ca$^{2+}$ permeability is dependent on this mode of activation, we repeated the ion substitution analysis in the presence of a photoactivatable Src kinase, PhoCl-Src, developed by the Thompson laboratory (University of Calgary, Calgary, AB, Canada) using technology created in collaboration with Dr R. Campbell (University of Alberta) (Zhang et al., 2017). Briefly, a construct was generated where the Src kinase domain was 'sandwiched' between two PhoCl and modified oestrogen receptors (ER). Here, ER dimerization sequestered the active kinase domain, protecting it from substrates prior to photocleavage. Following photocleavage of PhoCl, the Src kinase domain was released and acted on its substrate Panx1 Y308. To assess Panx1 Y308 phosphorylation with PhoCl-Src, HEK293T cells were seeded in 35 mm cell culture plates, then transiently transfected with rPanx1-myc ± PhoCl-Src for 48 h prior to the onset of illumination. Plates were then either sub-ject to 30 min of illumination with 365 nm light or placed in similar conditions in the absence of illumination, then returned to the cell culture incubator (37°C, 5% CO$_2$) for 1 h prior to generating whole cell lysates in NP40 lysis buffer and western blot analysis, as previously described (Weilinger et al., 2016).

To assess Ca$^{2+}$ permeability of Panx1 following PhoCl-Src activation, HEK293T cells were transiently transfected with rPanx1-MYC and PhoCl-Src (2.5:1 ratio) prior to photo-illumination (380 nm LED) and ion replacement (see above). Currents and Ca$^{2+}$ selectivity were recorded under continuous photo-illumination in the presence of high external Ca$^{2+}$.

### Antibodies

Mouse anti-myc, mouse anti-GAPDH and rabbit anti-pY416SFK primary antibodies were from Cell Signaling Technology (Danvers, MA, USA) (#2276, #2118L and #6943, respectively). Rabbit anti-Panx1 pY308 (pY308 Panx1) was designed in house and generated by New England Peptide (Gardner, MA, USA) and described by Weilinger et al. (2016).

Protein expression was detected using IRDye secondary antibodies (LiCOR, Lincoln, NE, USA) imaged on a Odyssey CLx Imaging system (LiCOR).

### Statistical analysis

Electrophysiological data were digitized at 10 kHz and low-pass filtered at 1 kHz using a MultiClamp 700B amplifier and Digidata 1440A analogue to digital converter (Molecular Devices, San Jose, CA, USA). Data were collected and stored in Clampfit 10.3/10.7 software (Molecular Devices) and analysed using Clampfit 10.3/10.7, Excel (Microsoft Corp., Red, pnd, WA, USA) and Prism, version 7 (GraphPad Software Inc., San Diego, CA, USA). Statistical analysis was determined using one-way ANOVA (comparisons of three or more groups) with a *post hoc* Tukey test or Student's *t* test (paired and unpaired). $P < 0.05$ was considered statistically significant, The results are presented as the mean ± SD. For LTD data, the last 5 min of baseline EPSC peak amplitude and the last 5 min of post-LFS EPSC peak amplitude were averaged. For analysis, cells were split into two groups based on the last 5 min peak average: (1) <85% of baseline or (2) >85% of baseline. Only cells that fell into the majority category for each condition were compared and analysed further. Cells were normalized both within individual cells and by group, depending on the down-stream analysis (indicated in graphs). The paired pulse ratio (PPR) was calculated by dividing the second EPSC peak by the first EPSC peak. The last 1 min baseline PPR average was used for comparison against the last 1 min post-LFS PPR average.

# Results

LTD in the hippocampus can be produced with 900 pulses of a LFS in the range 1–5 Hz (Dudek & Bear, 1992; Mulkey & Malenka, 1992). In 10 out of 14 cells, we observed a significant depression of EPSC peak amplitudes compared to baseline ($53.3 \pm 7.2\%$; paired Wilcoxon test, $P = 0.002$) of CA1 pyramidal neurons following 5 min (900 pulses) of 3 Hz stimulation to the Schaffer collaterals (Fig. 1*A*,*B*). EPSC peak depression was not a result of changes in presynaptic release probability because the PPR was not significantly altered (paired Wilcoxon test, $P = 0.4316$) following LFS (Fig. 1*B*). These results confirmed that our LTD protocol produced reliable and robust postsynaptic depression. All remaining experiments were conducted with a 3 Hz stimulation.

## Hippocampal LTD with transient NMDAR pore block is permissive for niNMDAR LTD

Despite increasing support for niNMDAR signalling (Birnbaum et al., 2015; Carter & Jahr, 2016; Kessels et al., 2013; Nabavi et al., 2013; Stein et al., 2015; Tamburri et al., 2013; Weilinger et al., 2016), its role in hippocampal LTD is controversial (Babiec et al., 2014; Nabavi et al., 2013). By contrast to niNMDAR LTD reported by Nabavi et al. (2013), Babiec et al. (2014) found that blocking ion flow through NMDARs with MK-801 completely attenuated LTD in mouse CA1 neurons. We first tested whether blocking ion conduction by NMDARs during and after LFS affected LTD. This was achieved by bath applying MK-801 (20 μм) 5 min prior to LFS and constantly during the recording period. The continuous presence of MK-801 prevented LTD induction and maintenance (Fig. 1*C*,*D*). The presence of MK-801 alone did not alter EPSC amplitude prior to LFS because amplitudes ($91.6\% \pm 2.1\%$ of control) were not significantly changed in the 5 min following MK-801 application ($P = 0.1875$, Wilcoxon test).

Activation of niNMDAR pathways during excitotoxicity is slow compared to synaptic responses, manifesting 5–10 min after the first NMDAR currents (Weilinger et al., 2012, 2016). We therefore considered whether transient inhibition of NMDARs altered the contribution of niNMDARs to synaptic strength. This was achieved by adding MK-801 10 min prior to and during LFS, but washing it out after synaptic stimulation. We have shown that this approach blocks synaptic responses but not niNMDAR signalling (Weilinger et al., 2016). The effect of transient MK-801 on LTD was time-dependent. In the first 5 min after LFS, transient MK-801 prevented depression of EPSCs compared to the control, drug-free condition (two-way ANOVA, $P = 0.0037$) (Figs 1*E*,*F* and 2*A*). Interestingly, EPSC amplitude in nine of 13 cells depressed gradually over time, so that it was $71.9 \pm 5.3\%$

of baseline at 20 min (two-way ANOVA, $P = 0.0046$). EPSCs were fully depressed by the last 5 min of the 40 min recording ($63.10 \pm 3.7\%$; two-way ANOVA, $P = 0.0002$) (Figs 1*E*,*F* and 2*A*,*B*). LTD in the last 5 min was not significantly different than the control group (two-way ANOVA, $P = 0.4960$) (Fig. 2*A*,*B*).

Inhibition by MK-801 probably persisted for the entire recording despite its washout from the bathing solution after LFS given its slow reversal time constant of pore block of ~90 min (Huettner & Bean, 1988). However, to rule out the possibility that a small population of NMDARs are unblocked and conducting and that this is responsible for the slowly developing LTD, we transiently blocked NMDARs with the competitive antagonist, D-APV (50 μм), prior to and during LFS. D-APV was washed from the bath after LFS, similar to transient MK-801 application. In six of eight cells, D-APV prevented depression of the EPSC peak amplitude from the baseline ($103.19 \pm 8.0\%$; paired Wilcoxon test, $P = 0.6875$) following LFS (Fig. 1*G*,*H*). This was not a result of presynaptic effects of D-APV because the PPR remained unchanged (paired Wilcoxon test, $P = 0.3125$) (Fig. 1*H*). D-APV by itself did not alter EPSC amplitude, which was $114.6 \pm 20.1\%$ of the control value (paired Wilcoxon test, $P = 0.0625$), 5 min after application and prior to LFS. Compared to control conditions, preventing glutamate binding to NMDARs with D-APV completely attenuated hippocampal LTD throughout the 40 min recording period (Kruskal–Wallis test, $P < 0.001$) (Fig. 2*A*). LTD with transient MK-801 was statistically different than that with transient D-APV at 20 min (Kruskal–Wallis test, $P = 0.0109$) (Fig. 2*A*). EPSC were fully depressed by the last 5 min of the 40 min recording following transient MK-801 application ($63.10 \pm 3.7\%$; paired Wilcoxon, $P = 0.0039$) (Figs 1*E*,*F* and 2*A*,*B*). Together, these data reveal that persistent inhibition of NMDARs by MK-801 blocks LTD, but transient inhibition during the induction phase is permissive for a niNMDAR contribution to LTD unless glutamate binding is inhibited by D-APV.

## Panx1 is required for LTD

Panx1 is involved in synaptic plasticity (Ardiles et al., 2014; Gajardo et al., 2018) and we predicted that it could be a downstream target of niNMDARs during LTD. To selectively block Panx1, we used the peptide inhibitor, [10]panx (WRQAAFVDSY; 100 μм) or its scrambled control, [SC]panx, (FSVYWAQADR; 100 μм). Bath application of [10]panx was prior to and during LFS but washed away following LFS. The Panx1 blocker did not by itself alter EPSC amplitude ($91.1\% \pm 4.5\%$ of control; Wilcoxon, $P = 0.1875$) but blocked LFS-induced LTD, and unmasked a slight, but non-significant potentiation

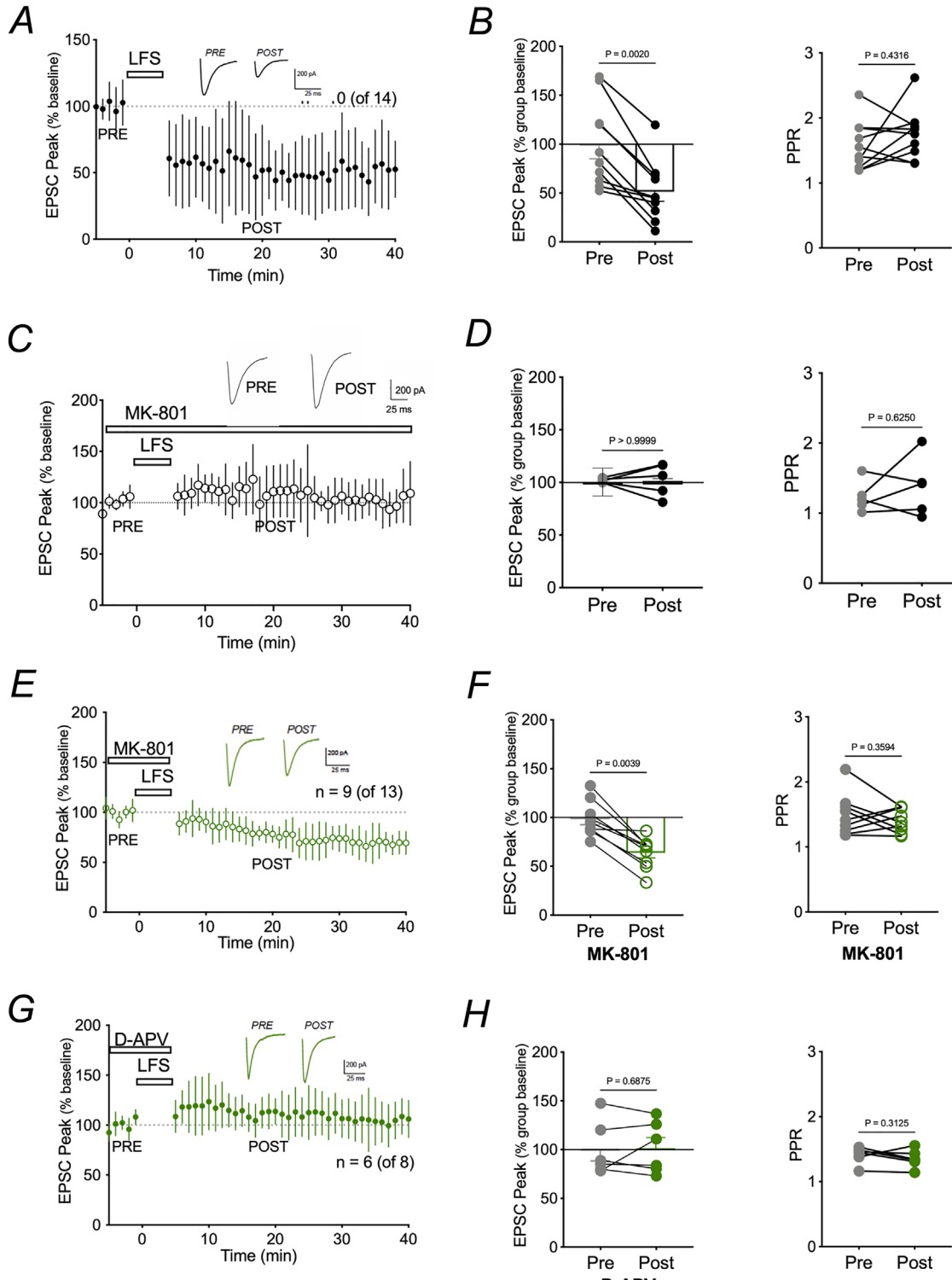

**Figure 1. Differential effects of transient MK-801 and D-APV on 3 Hz low frequency stimulation-dependent LTD**

*A*) mean ± SD of peak EPSC amplitude and representative traces (inset) for pre- and post 3 Hz stimulation of the Shaffer collaterals showing robust LTD. Dashed line represents 100% of baseline. *B*) left: mean peak EPSC amplitudes pre- (grey, 5 min average) and post- (black, 5 min average) 3 Hz LFS. Note the robust depression in amplitude. Data were normalized to the mean of the entire dataset. Right: paired pulse ratio (PPR) was unchanged following 3 Hz LF). *C*) mean ± SD of peak EPSC amplitude in the continual presence of MK-801. Note that LTD was inhibited. *D*) left: mean peak EPSC amplitudes pre and post LFS. Right: PPR. Both were unchanged in the

presence of MK-801. *E)* mean ± SD of peak EPSC amplitude and representative traces (inset) pre- and post-LFS in the presence of 20 µM MK-801. Note that MK-801 was present during the pre- and LFS times but washed out following LFS of Shaffer collaterals. *F)* EPSC amplitude was significantly depressed (Student's *t* test, *P* = 0.0002, *n* = 9) in the presence of transient 20 µM MK-801, but PPR (right panel) remained unchanged (Student's *t* test, *P* = 0.4356, *n* = 9). *G)* mean ± SD of peak EPSC amplitude and representative traces (inset) pre- and post-LFS in the presence of 50 µM D-APV. Note that D-APV was present during the pre- and LFS times but washed out following LFS of Shaffer collaterals. *H)* peak EPSC amplitude pre- and post-LFS (left) in the presence of D-APV. EPSC amplitude was not significantly depressed by µM D-APV. Right: PPR both pre-and post-LFS with D-APV present in the pre- and LFS times.

of EPSCs from baseline (128.66 ± 7.8%; Wilcoxon, *P* = 0.0625) in seven out of 11 cells (Fig. 3*A,B*). LFS-induced LTD in the presence of [SC]panx was not different than the control (i.e. Fig. 3*D* compared to Fig. 1*B*; Kruskal–Wallis, *P* > 0.999) and produced significant depression (52.42 ± 9.3%; Kruskal-Wallis, *P* < 0.0069) in six out of six cells (Fig. 3*C,D*). Neither [10]panx, nor [SC]panx produced significant changes in PPR (Fig. 3*B,D*). These data support a role for postsynaptic Panx1 in LTD.

### niNMDAR to Panx1 pathways contribute to LTD

Opening of Panx1 occurs during ischaemia by niNMDAR activation of Src kinase (Weilinger et al., 2012, 2016). We therefore tested whether blocking Src could prevent LTD. Src was blocked with PP2 (1 µM) included in the patch pipette, which by itself did not alter EPSC amplitude (87.0 ± 6.8% of baseline, Wilcoxon, *P* = 0.0625). The inactive enantiomer of PP2, PP3 (1 µM) was used as a control. Inhibition of Src with PP2 prevented LTD in four of six

cells (119.4 ± 11.6%; Wilcoxon, *P* = 0.875) (Fig. 4*A,B*). By contrast, LTD was not affected by PP3; EPSCs were significantly reduced (67.99 ± 6.6% of the PP2 level; Wilcoxon, *P* = 0.0312) in six out of 10 cells (Fig. 4*C,D*). LTD in the presence of PP3 was not statistically different than control (Kruskal–Wallis, *P* = 0.7895) (Fig. 5*G*). Thus, LTD required both Panx1 (above) and Src.

Src kinase activation of Panx1 can be disrupted during ischaemia with the interfering peptide, TAT-Panx[Y308] (TAT-LKVYEILPTFDVLH), which prevents Panx1 phosphorylation (Weilinger et al., 2016). To further support the interplay between NMDARs, Src kinase and Panx1 during LTD, we bath applied the cell permeable TAT-Panx[Y308] (1 µM), which did not alter pre-LFS EPSC amplitude (109.4% ± 13.3% of baseline, Wilcoxon, *P* = 0.125) but prevented LTD and promoted an LTP-like response in five out of eight cells where EPSC were 156.7% ± 19.9% of baseline following LFS (Kruskal–Wallis, *P* = 0.0049) (Fig. 5*C*). Compared to the scrambled control (TAT-SC[Y308], 1 µM), EPSCs were 53.48% ± 12.2% of the baseline (Kruskal–Wallis *P* = 0.0204) (Fig. 5*C,D*).

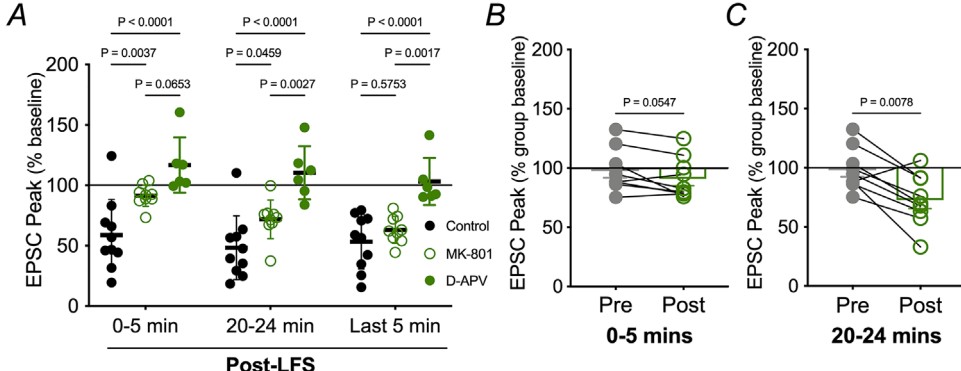

**Figure 2. Summary of the time-course effects of different transiently applied NMDAR blockers on LTD**
*A)* EPSC amplitude averaged from 0 to 5 min and 20–24 min after 3 Hz LFS of the Shaffer collaterals. Comparisons were made with a two-way ANOVA (*F* = 14.27, d.f. = 22) that had significant source of variation for drug (4.597%, *P* = 0.0319) and time (51.32%, *P* < 0.0001). Note that the competitive glutamate site antagonist, 50 µM D-APV (*n* = 6) significantly blocked LTD compared to control throughout the recording period. By contrast, transient MK-801 (*n* = 9) prevented EPSC depression in the first 5 min after LFS; amplitudes were significantly different than those in the presence of D-APV but were not significantly different than control. By 20 min, EPSC amplitudes in the presence of MK-801 were different than both the control and D-APV, and, in the last 5 min of the recording time, EPSC amplitude was depressed to a level similar to the control and were different than D-APV at the same time. *B)* comparison of the peak EPSC amplitude for individual cells before LFS and after MK-801 plus LFS. Amplitudes were not significantly different. *C)* comparison of peak EPSC amplitudes 25 min after LFS in the presence of transient MK-801. EPSC amplitude was significantly suppressed. Note that MK-801 was washed out immediately following LFS.

LTD in the presence of TAT-SC$_{Y308}$ was not statistically different than in its absence (Kruskal–Wallis, $P > 0.9999$) in six out of eight cells (Fig. 5*C*). Together, these data support a role for niNMDAR signalling to Panx1 through Src kinase in 3 Hz LFS LTD.

## Lack of calcium permeability of Panx1

We hypothesized that Panx1 may serve as an alternative source of Ca$^{2+}$ entry to NMDARs. To test this, we first evaluated whether voltage activated Panx1 is Ca$^{2+}$ permeable because some studies have suggested it is not (Ma et al., 2012). To this end, we conducted ion replacement studies in HEK-293T cells co-transfected with GFP and rat Panx1 (1:5 ratio). GFP-positive cells were voltage clamped and the membrane potential ramped from $-80$ to $+80$ mV to activate Panx1 and generate *I–V* curves under control Ca$^{2+}$ (2 mM) and high Ca$^{2+}$ (40 mM) conditions. *I–V* curves were used to determine the reversal potential ($E_{rev}$). The E$_{rev}$ under control conditions (2 mM Ca$^{2+}$) was $-8.0 \pm 0.6$ mV. When extracellular Ca$^{2+}$ was increased 20-fold, $E_{rev}$ was $7.5 \pm 0.5$ mV. This was not significantly different from 2 mM

Ca$^{2+}$ (paired *t* test, $P = 0.2324$), suggesting that Panx1 is not directly permeable to Ca$^{2+}$ (Fig. 6*A,B*). To confirm our ability to detect significant shifts in the $E_{rev}$ of Panx1, we replaced 140 mM NaCl with 140 mM NaGluc. Panx1 permeability to Cl$^-$ has been described (Ma et al., 2012; Romanov et al., 2012; Wang et al., 2014). Reducing Cl$^-$ concentrations significantly shifted the $E_{rev}$ of Panx1 to $41.1 \pm 0.9$ mV (paired *t* test, $P < 0.0001$) (Fig. 6*C,D*).

Panx1 functions in multiple conductance modes, where the smaller conductance state is assumed to be predominantly Cl$^-$ permeable and the large conductance state is permeable to ATP and Ca$^{2+}$ (Wang et al., 2018). Our work demonstrated that the large pore state is activated by niNMDAR pathways during ischaemia and cell death via phosphorylation of Panx1 at Y308 by Src (Weilinger et al., 2016). To mimic Src phosphorylation of Panx1, we took advantage of a photocleavable protein, PhoCl, engineered to release target proteins upon ultraviolet illumination (Zhang et al., 2017). Here, we generated a PhoCl-Src, where a rat Src kinase domain was inserted between two PhoCl proteins, bookended by modified estrogen receptors (Fig. 6*E*). ER dimerization masked Src kinase activity. In the presence of ultraviolet

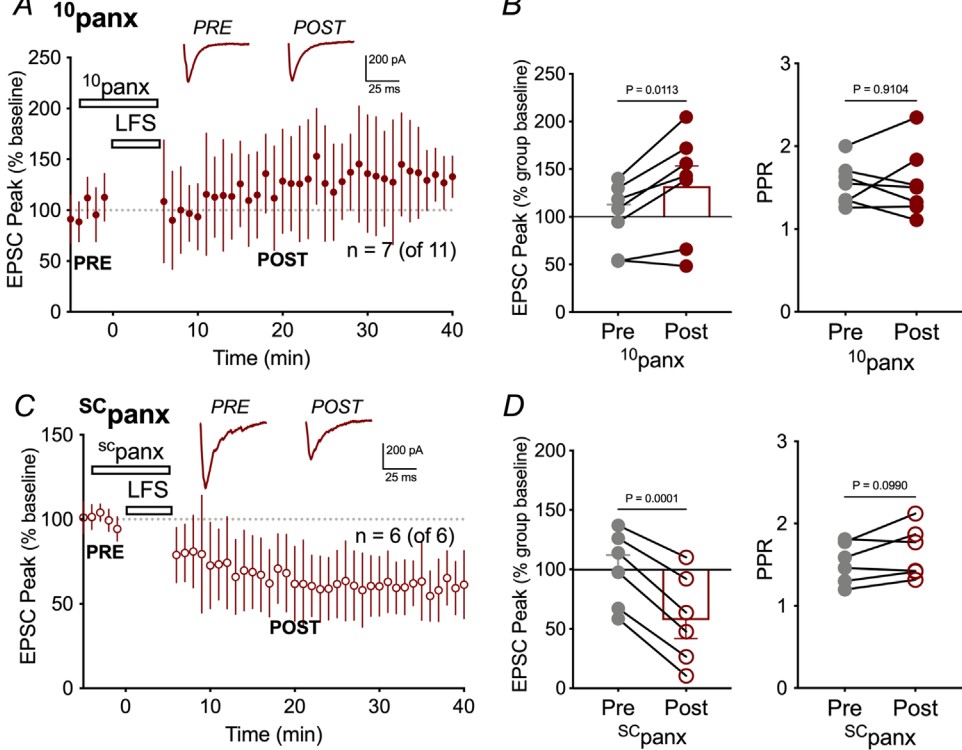

**Figure 3. Panx1 is required for LTD**

*A*) mean $\pm$ SD of EPSC amplitude over time with representative traces pre- and post-LFS for $^{10}$panx as insets. *B*) EPSC amplitude showed non-significant potentiation with 100 µM $^{10}$panx in the bath. Right: comparing PPR for individual neurons. Note no significant change. *C*) mean $\pm$ SD of EPSC amplitude over time with representative traces pre- and post-LFS for $^{sc}$panx, the scrambled control for $^{10}$panx as insets. *D*) left: comparing EPSC peak amplitude for individual neurons in the presence of $^{sc}$panx, which was significantly depressed with 100 µM $^{sc}$panx in six of six cells. Right: PPR was unchanged by $^{sc}$panx.

light (i.e. 365–380 nm), PhoCl domains undergo a peptide backbone cleavage to release the active kinase domain (Fig. 6*E*). We then expressed rat Panx1 and PhoCl-Src in HEK 293T cells to assess phosphorylation of Panx1 at Y308 (Fig. 6*F*) and determine whether Src activation of Panx1 altered $Ca^{2+}$ permeability.

To assess Panx1 phosphorylation, HEK 293T cells were transfected with rat Panx1-myc with or without co-transfection of PhoCl-Src. After 48 h, plated cells were either placed in a UV illumination chamber for 30 min (365 nm) or in a similar environment absent of illumination. Next, cells were returned to the incubator for 1 h prior to the generation of whole cell lysates and western blot analysis. PhoCl-Src activation via photocleavage increased phosphorylation at Panx1 Y308 (Fig. 6*F*).

To determine whether Src phosphorylation induced $Ca^{2+}$ permeability of Panx1, PhoCl-Src was stimulated with 380 nm LED light for an average of $10.6 \pm 3.0$ min when HEK 293T cells were voltage clamped and the membrane potential ramped from $-80$ to $+80$ mV. After activation of PhoCl-Src, the bathing solution was changed to one with 40 mm extracellular $Ca^{2+}$ and

$E_{\text{rev}}$ was determined. Despite an increase in whole cell current ($227.8\% \pm 41.9\%$ baseline; $P = 0.0221$ with Dunnett's multiple comparison *post hoc* test, $n = 6$) (Fig. 6*G*), no change in the $E_{\text{rev}}$ of Panx1 was detected ($E_{\text{rev}} = -7.0 \pm 1.2$ mV; $P = 0.4037$ compared to control with Dunnett's multiple comparison *post hoc* test, $n = 6$) (Fig. 6*F–H*). This indicates that Panx1 activation, either by voltage alone or in conjunction with Src-dependent Y308 phosphorylation, does not produce $Ca^{2+}$ permeability, suggesting that there is an alternative explanation for the role of Panx1 in LTD.

## ATP release through Panx1 may drive LTD by recruiting P2X4 receptors

The lack of $Ca^{2+}$ permeability of Panx1 suggests it is not directly involved in increasing $Ca^{2+}$ that may be required for LTD. This led us to investigate an alternative role for Panx1, based on the channel's extensive characterization as an ATP-release channel (Bao et al., 2004; Chiu et al., 2018; Dahl, 2015; Locovei, Bao et al., 2006; Qu et al., 2011) that has been linked to the activation of P2Y (Locovei, Bao

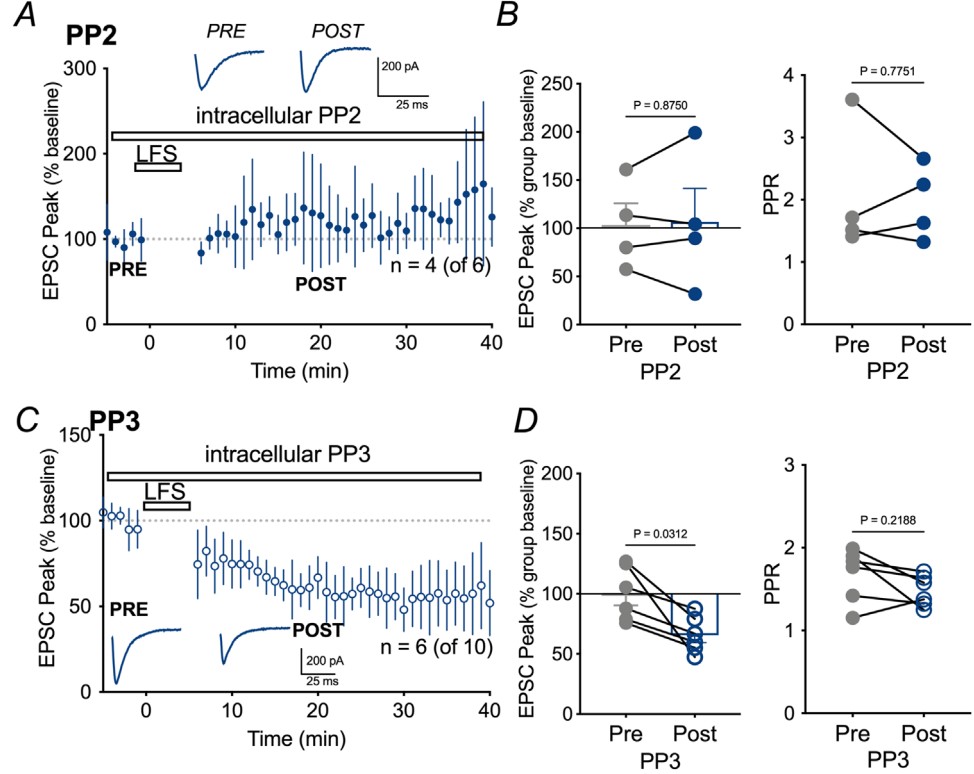

**Figure 4. Src family kinases are necessary for LTD**
*A*) mean $\pm$ SD of peak EPSC amplitude over time in the presence of the Src inhibitor, PP2 included in the patch pipette. Representative traces pre- and post-LFS for PP2 are shown as insets. *B*) left: no significant depression of EPSC amplitude occurred following LFS when PP2 was present. Right: PPR was unchanged by PP2. *C*) mean $\pm$ SD of peak EPSC amplitude over time in the presence of the PP2 control, PP3 included in the patch pipette. Representative traces pre- and post-LFS for PP3 are shown as insets. *D*) left: EPSC peak amplitude was significantly depressed in the presence of 1 μm PP3. Right: no change in PPR in the presence of PP3.

et al., 2006; Locovei, Wang et al., 2006) and P2X (Locovei et al., 2007; Maslieieva & Thompson, 2014) receptors. Application of ATP alone can induce synaptic depression (Pougnet et al., 2016; Yamazaki et al., 2003). We hypothesized that Panx1 may release ATP and activate downstream purinergic signalling to induce LTD.

To test this, we first examined whether exogenous ATP application could rescue TAT-Panx$_{Y308}$-blocked LTD. TAT-Panx$_{Y308}$ was perfused during baseline and remained present during the LFS when ATP (50 μM) was also added to the bath. Both drugs were washed out after LFS and EPSC peaks were analysed. ATP rescued LTD in the presence of TAT-Panx$_{Y308}$ and produced a significant depression in EPSC peak from baseline (53.80% ± 6.8%; Wilcoxon, $P = 0.0003$) in nine out of 10 cells (Fig. 7A,B). EPSC depression with ATP was not significantly different than the drug free control conditions (i.e. *vs.* Fig. 1A, Wilcoxon, $P = 0.9632$). ATP was probably not affecting presynaptic glutamate release, at least in the later stages, because the PPR at 20–25 min was not significantly altered from baseline PPR (Wilcoxon, $P = 0.0973$) (Fig. 7B).

We then evaluated the effects of ATP in the presence of D-APV on evoked synaptic responses without LFS because it is possible that ATP is depolarizing neurons leading to activation of NMDARs and subsequent LTD. If this is the case, we predicted that D-APV would prevent ATP induced depression of EPSCs. D-APV did not prevent depression of EPSCs by ATP alone (29.8% ± 17.8% of control; Wilcoxon $P < 0.0001$) (Fig. 7C,D), indicating that activation of purinergic receptors is sufficient to induce LTD even when both ionotropic and niNMDARs are blocked. As in all other experiments presented above, the PPR was not changed by ATP (Fig. 7D).

Recovery of LTD with exogenous ATP application suggests the involvement of purinergic receptors. Previous work in our laboratory found a link between P2X4 receptors (P2X4Rs) and Panx1 channels in the choroid plexus (Maslieieva & Thompson, 2014). We tested this possibility by blocking P2X4Rs with the specific antagonist, 5-BDBD (10 μM) added to the bath throughout the experiment. Inhibiting P2X4Rs prevented LFS induced LTD (98.69 ± 4.5%; Wilcoxon, $P = 0.9966$)

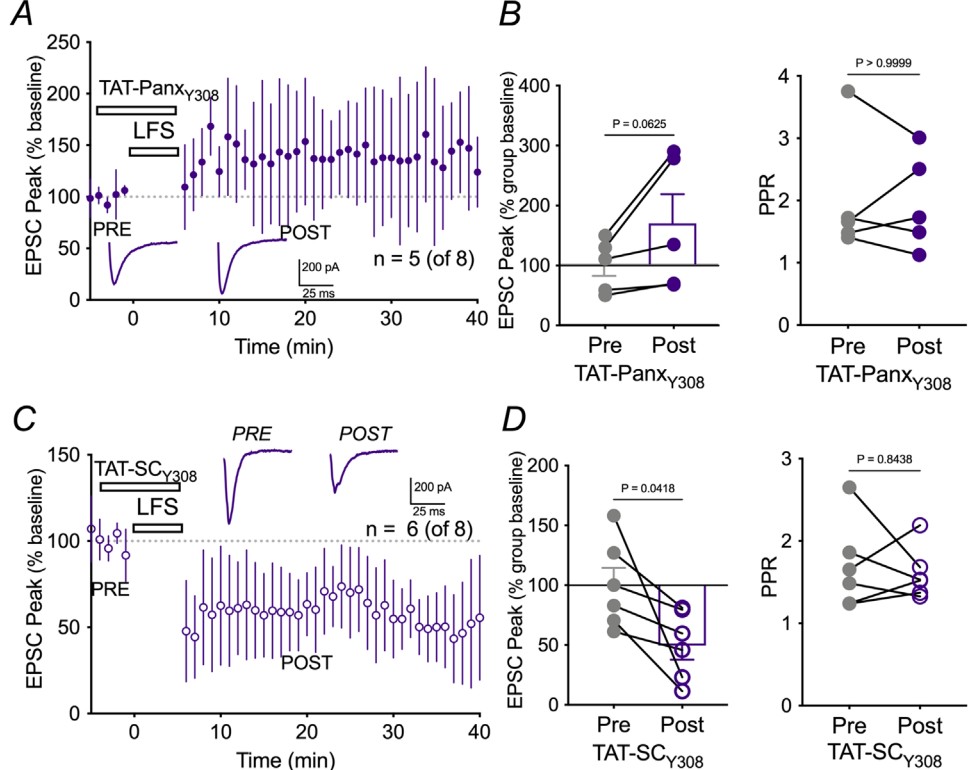

**Figure 5. LTD is blocked by an interfering peptide that prevents Src interaction with Y308 in the C-tail of Panx1**
*A)* mean ± SD of peak EPSC amplitude over time in the transient presence of the Src-Panx1 interaction inhibitor, TAT-Panx$_{Y308}$. Representative traces pre- and post-LFS for TAT-Panx$_{Y308}$ are shown as insets. *B)* left: no significant depression of EPSC amplitude occurred following LFS when TAT-Panx$_{Y308}$ was present. Right: PPR was unchanged by TAT-Panx$_{Y308}$. *C)* mean ± SD of peak EPSC amplitude over time in the transient presence of the TAT-Panx$_{Y308}$ control, TAT-SC$_{Y308}$. Representative traces pre- and post-LFS for TAT-SC$_{Y308}$ are shown as insets. *D)* left: EPSC peak amplitude was significantly depressed in the presence of TAT-SC$_{Y308}$. Right: no change in PPR in the presence of TAT-SC$_{Y308}$.

in five out of nine cells (Fig. 8*A*,*B*). This was not a result of changes in presynaptic probability of release because PPR was not significantly altered (Wilcoxon, $P = 0.3551$) (Fig. 8*C*). Comparison to control conditions (drug free) revealed a significant block in LTD (unpaired *t* test, $P = 0.0010$) (Fig. 8*D*). These data indicate that postsynaptic P2X4Rs are a necessary component of hippocampal LTD and may be activated by ATP released through Panx1.

## Discussion

In the present study, we have investigated a role for niNMDARs and Panx1 in LTD in the hippocampus. We report that transient block of ionotropic NMDARs with the pore blocker, MK-801, results in a slowly developing LTD. By contrast, transient block of NMDARs with the glutamate site competitive antagonist, D-APV, prevented LTD. The difference between transient application of D-APV (no-LTD) and MK-801 (delayed LTD) indicates that glutamate binding to the NMDAR during LFS of the CA3–CA1 synapse can activate niNMDAR pathways and induce LTD. Interestingly, the continuous presence

of MK-801 during the recording prevented LTD. This suggests that ionotropic NMDAR LTD is the predominant form at the CA3–CA1 hippocampus, although when conduction by NMDARs is blocked with MK-801 but glutamate binding is permitted during LFS, a niNMDAR mechanism involving Panx1 is activated, leading to slow LTD induction. Our data support the proposal that this is a *bona fide* niNMDAR LTD because, first, transient D-APV prevented LTD and given its fast dissociation rate of $0.13 \text{ s}^{-1}$ (Monaghan et al., 1984), indicates that glutamate binding during the LFS is critical for LTD induction. Second, the slow reversal of block of MK-801 of ~90 min (Huettner & Bean, 1988) suggests that the majority of NMDARs will not be conducting during the recording time of 40 min. These receptors would have bound glutamate during the LFS period, which appears sufficient to activate niNMDAR pathways. Our model suggests a novel role for the niNMDAR-Src-Panx1 signalling in LTD. We further provide evidence supporting ATP release and recruitment of downstream P2X4 purinergic receptors in niNMDAR LTD.

It is important to address the activity-dependent nature of MK-801 block of NMDARs (Huettner &

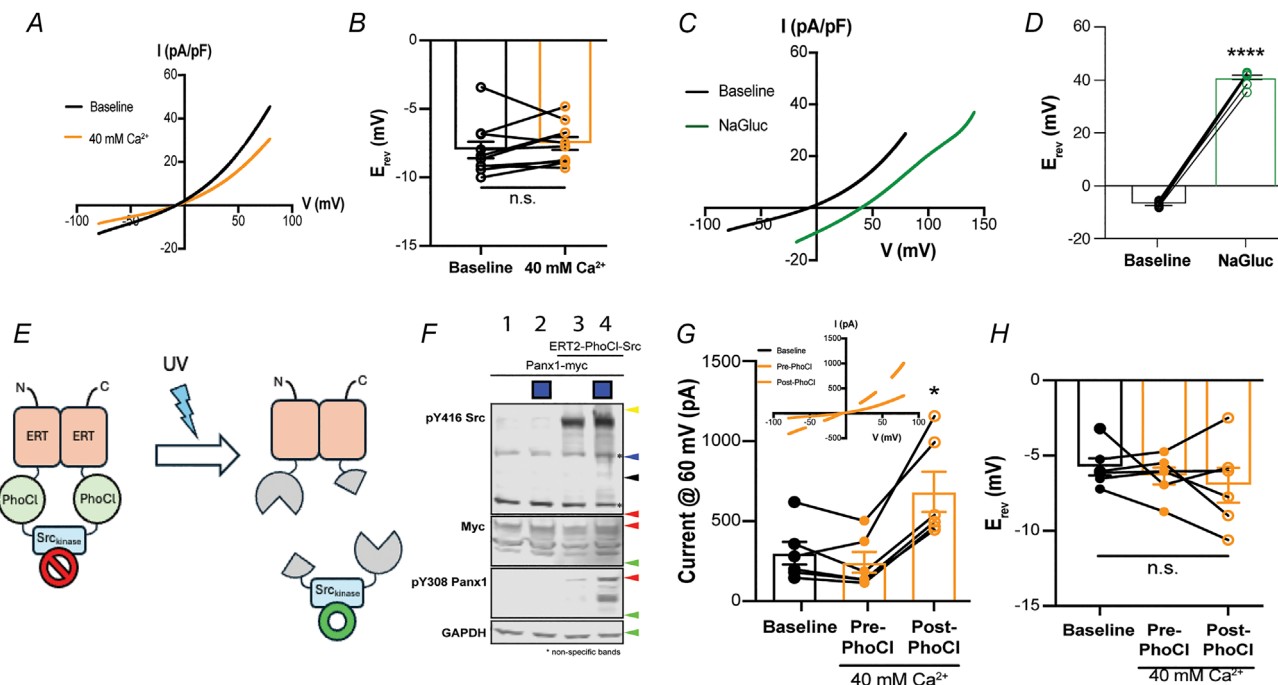

**Figure 6. Panx1 is not directly $Ca^{2+}$ permeable, even when activated by Src kinase**
*A*) current–voltage ramps of HEK 293T cells expressing rat Panx1 and GFP with 2mM or 40 mM $CaCl_2$ in the bathing solution. *B*) comparison of whole-cell reversal potential in 2 and 40 mM extracellular $CaCl_2$. *C*) current–voltage ramps of HEK 293T cells expressing rat Panx1 and GFP when extracellular NaCl is replaced by Na-gluconate (NaGlu). Note the prominent rightward shift. *D*) replacement of extracellular NaCl with NaGlu significantly changed whole-cell reversal potential, indicating that $Cl^-$ is a Panx1 permeant. *E*) model of the activation of PhoCl-Src. *F*) western blot showing that 365 nm UV illumination effectively activated Src and led to phosphorylation of Panx1 at Y308. *G*) inset: current–voltage ramps of HEK 293T cells expressing rat Panx1 and PhoCl-Src. Stimulation of PhoCl-Src with 380 nm illumination increases whole-cell Panx1 currents. *H*) comparison of the reversal potential after PhoCl-Src activation of Panx1 does not induce significant $Ca^{2+}$ permeability.

Bean, 1988). As previously reported, we found NMDAR current inhibition within 10 min of MK-801 perfusion with concurrent synaptic stimulation (Weilinger et al., 2016). Other studies have incubated for multiple hours to ensure sufficient NMDAR block (Babiec et al., 2014; Nabavi et al., 2013). To verify our effect was a result of niNMDAR signalling and not incomplete inhibition, we utilized other pharmacological methods to test downstream components of the niNMDAR-Src-Panx1 pathway. The ability of [10]panx, PP2 and TAT-Panx$_{Y308}$ to prevent LTD when applied during LFS provides confidence that the effect we are seeing is the result of an niNMDAR pathway and probably not delayed reversal of MK-801 inhibition. The washout period following LFS led to gradual depression of EPSCs over 20 min post-stimulation (when MK-801 is no longer in the bath). During this time, glutamate levels would be expected to return to basal concentrations because LFS is no longer present. This suggests that there is a window for ligand binding to NMDARs that activates niNMDAR signalling, and we propose that this causes ATP release, leading to slow LTD via recruitment of P2X4.

Panx1 was not $Ca^{2+}$ permeable when expressed in HEK 292T cells, suggesting it is probably not support $Ca^{2+}$ influx into neurons during LTD. We were unable to alter $Ca^{2+}$ permeability by direct activation of PhoCl-Src, which increased Panx1 currents. This prompted us to investigate a role for Panx1 in ATP release and downstream purinergic signalling pathways in LTD. We report that ATP can rescue LFS induced LTD when Panx1 was blocked during LFS, albeit with different kinetics that are most probably a result of bath application of ATP that could recruit a larger variety of purinergic receptors compared to focal release from the cell. Additionally, ATP in the presence of D-APV to block both ionotropic and niNMDARs evoked significant depression of EPSCs in the absence of LFS, indicating that purinergic receptors alone can cause LTD. ATP can be released from multiple sources, including presynaptic nerve terminals, astrocytes and large pore channels, including Panx1 (Araque et al., 2014; Bao et al., 2004; Burnstock, 2007; Dahl, 2015; Lalo et al., 2014). The ability of 5-BDBD to block LTD in the present study supports the involvement of ATP and P2X4R. We are unclear of the final source of ATP to

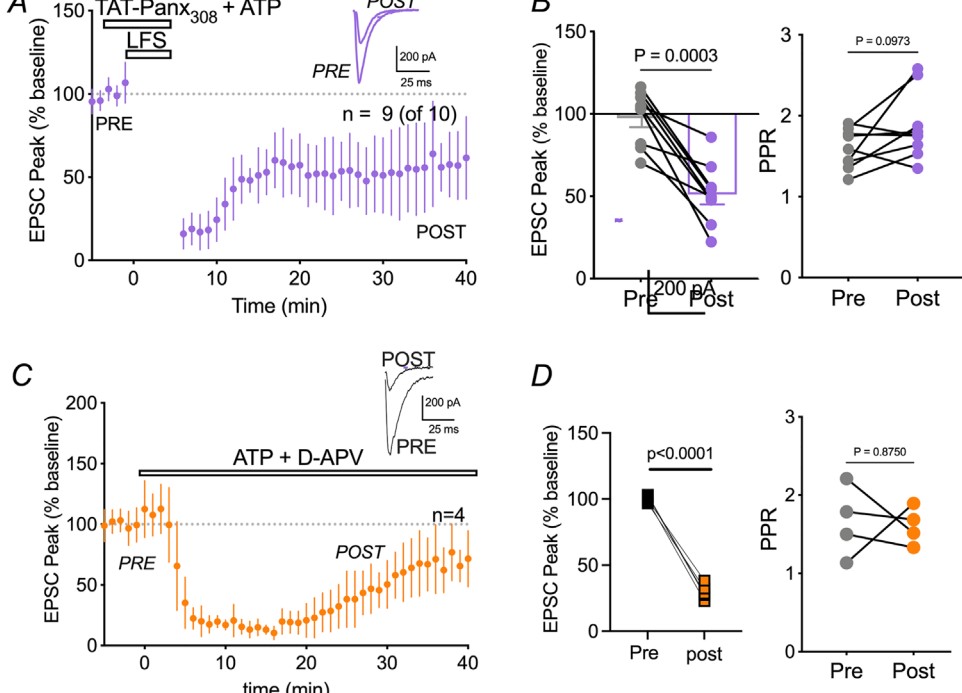

**Figure 7. Exogenous ATP rescues TAT-Panx$_{Y308}$-blocked LTD**
*A*) mean ± SD over time with representative traces (inset) for induction of LTD in the presence of TAT-Panx$_{Y308}$ + 50 μM ATP. Dashed line represents 100% of baseline. *B*) average peak amplitudes pre-LFS (last 5 min) and post-LFS (last 5 min). Application of ATP produced a depression in EPSC peak amplitude from baseline. Right: PPR was unchanged in the last 1 min of post-LFS recording compared to the last 1 min of baseline. *C*) EPSCs were significantly depressed by ATP in the absence of 3 Hz frequency synaptic stimulation and the presence of ATP. *D*) comparison of peak EPSC amplitudes in the presence of ATP + D-APV but without 3 Hz synaptic stimulation shows significant depression. The PPR was unchanged. This suggests that ATP is sufficient to induce LTD without NMDAR activity.

induce LTD following LFS. It is possible that neuronal Panx1 releases factors that promote ATP release from astrocytes, such as glutamate or other transmitters. It has been reported that astrocytic ATP can cause robust LTD (Zhang et al., 2003) and this could account for the slow LTD when in the presence of transient MK-801.

In conclusion, we present data suggesting that niNMDARs are activated by 3Hz LFS to induce LTD at the CA3–CA1 synapse when NMDARs are transiently blocked by MK-801, but not D-APV. When NMDARs are blocked by MK-801 throughout the recording period, ion flux through NMDARs appears to be critical for synaptic depression. The niNMDARs activate Src and subsequently Panx1, which we propose releases ATP to recruit P2X4Rs and induce LTD. It is not yet known whether P2X4 allows $Ca^{2+}$ influx to influence post-synaptic responses, but they do not appear to induce depolarization of the membrane to activate ionotropic NMDARs because D-APV prevented the depression of EPSCs by ATP alone.

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

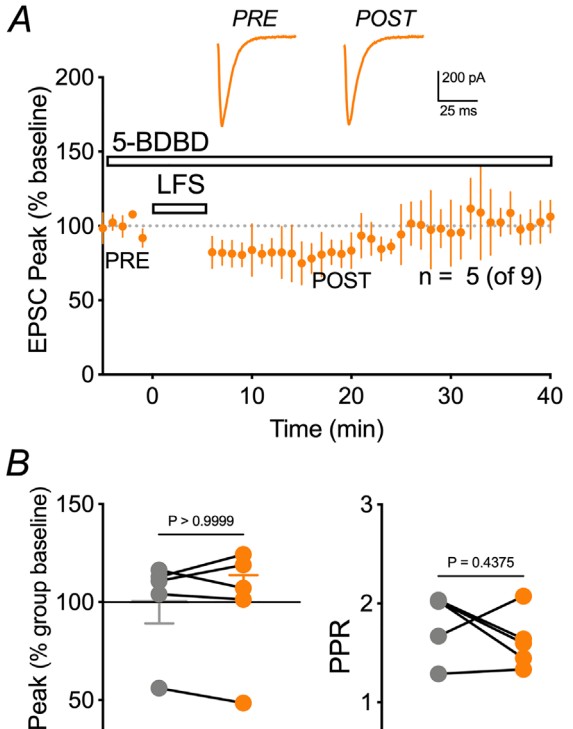

**Figure 8. Inhibiting P2X4Rs significantly attenuates LFS-induced LTD**

*A*) mean ± SD time-course of peak EPSC amplitude, with representative traces (inset) for 3 Hz LFS induced LTD in the presence of the P2X4R blocker, 10 μM 5-BDBD. Dashed line represents 100% of baseline. *B*) left: peak amplitudes for individual neurons pre-LFS (last 5 min) and post-LFS (last 5 min). LFS caused no significant depression from baseline following LFS when P2X4Rs were blocked. Right: PPR was unaffected. Thus, 5-BDBD significantly attenuated LTD compared to control cells, supporting a role for P2X4Rs in LFS-induced LTD.

Chater, T. E., & Goda, Y. (2014). The role of AMPA receptors in postsynaptic mechanisms of synaptic plasticity. *Frontiers in Cellular Neuroscience*, **8**, 401.

Chiu, Y.-H., Schappe, M. S., Desai, B. N., & Bayliss, D. A. (2018). Revisiting multimodal activation and channel properties of Pannexin 1. *Journal of General Physiology*, **150**(1), 19–39.

Chowdhury, D., & Hell, J. W. (2018). Homeostatic synaptic scaling: Molecular regulators of synaptic AMPA-type glutamate receptors. *F1000Research*, **7**, 234.

Citri, A., & Malenka, R. C. (2008). Synaptic plasticity: Multiple forms, functions, and mechanisms | neuro-psychopharmacology. *Neuropsychopharmacology (New York, NY)*, **33**(1), 18–41.

Dahl, G. (2015). ATP release through pannexon channels. *Philosophical Transactions of the Royal Society of London Series B, Biological Sciences*, **370**(1672), 20140191.

Derkach, V., Barria, A., & Soderling, T. R. (1999). $Ca^{2+}$/calmodulin-kinase II enhances channel conductance of $\alpha$-amino-3-hydroxy-5-methyl-4-isoxazolepropionate type glutamate receptors. *Proceedings of the National Academy of Sciences*, **96**(6), 3269–3274.

Dore, K., Aow, J., & Malinow, R. (2015). Agonist binding to the NMDA receptor drives movement of its cytoplasmic domain without ion flow. *Proceedings of the National Academy of Sciences of the United States of America*, **112**(47), 14705–14710.

Dudek, S. M., & Bear, M. F. (1992). Homosynaptic long-term depression in area CA1 of hippocampus and effects of N-Methyl-D-aspartate receptor blockade. *Proceedings of the National Academy of Sciences of the United States of America*, **89**(10), 4363–4367.

Gajardo, I., Salazar, C. S., Lopez-Espíndola, D., Estay, C., Flores-Muñoz, C., Elgueta, C., Gonzalez-Jamett, A. M., Martínez, A. D., Muñoz, P., & Ardiles, Á. O. (2018). Lack of Pannexin 1 alters synaptic GluN2 subunit composition and spatial reversal learning in mice. *Frontiers in Molecular Neuroscience*, **11**, 114.

Goda, Y., & Stevens, C. F. (1996). Synaptic plasticity: The basis of particular types of learning. *Current Biology*, **6**(4), 375–378.

Gray, J. A., Zito, K., & Hell, J. W. (2016). Non-ionotropic signaling by the NMDA receptor: Controversy and opportunity. *F1000Research*, **5**, F1000.

Ho, V. M., Lee, J.-A., & Martin, K. C. (2011). The cell biology of synaptic plasticity. *Science (New York)*, **334**(6056), 623–628.

Huettner, J. E., & Bean, B. P. (1988). Block of N-methyl-D-aspartate-activated current by the anticonvulsant MK-801: Selective binding to open channels. *Proceedings of the National Academy of Sciences of the United States of America*, **85**(4), 1307–1311.

Kessels, H. W., Nabavi, S., & Malinow, R. (2013). Metabotropic NMDA receptor function is required for $\beta$-amyloid-induced synaptic depression. *Proceedings of the National Academy of Sciences of the United States of America*, **110**(10), 4033–4038.

Lalo, U., Palygin, O., Rasooli-Nejad, S., Andrew, J., Haydon, P. G., & Pankratov, Y. (2014). Exocytosis of ATP from astrocytes modulates phasic and tonic inhibition in the neocortex. *PLoS Biology*, **12**(1), e1001747.

Locovei, S., Bao, L., & Dahl, G. (2006). Pannexin 1 in erythrocytes: Function without a gap. *Proceedings of the National Academy of Sciences*, **103**(20), 7655–7659.

Locovei, S., Scemes, E., Qiu, F., Spray, D. C., & Dahl, G. (2007). Pannexin1 is part of the pore forming unit of the P2X(7) receptor death complex. *Federation of European Biochemical Societies Letters*, **581**(3), 483–488.

Locovei, S., Wang, J., & Dahl, G. (2006). Activation of pannexin 1 channels by ATP through P2Y receptors and by cytoplasmic calcium. *Federation of European Biochemical Societies Letters*, **580**(1), 239–244.

Ma, W., Compan, V., Zheng, W., Martin, E., North, R. A., Verkhratsky, A., & Surprenant, A. (2012). Pannexin 1 forms an anion-selective channel. *Pflugers Archiv: European Journal of Physiology*, **463**(4), 585–592.

Malenka, R. C. (1994). Synaptic plasticity in the hippocampus: LTP and LTD. *Cell*, **78**, 535–538.

Malenka, R. C., & Bear, M. F. (2004). LTP and LTD: An embarrassment of riches. *Neuron*, **44**(1), 5–21.

Mammen, A. L., Kameyama, K., Roche, K. W., & Huganir, R. L. (1997). Phosphorylation of the $\alpha$-Amino-3-hydroxy-5-methylisoxazole4-propionic acid receptor GluR1 subunit by calcium/calmodulin-dependent Kinase II. *Journal of Biological Chemistry*, **272**, 32528–32533.

Maslieieva, V., & Thompson, R. J. (2014). A critical role for pannexin-1 in activation of innate immune cells of the choroid plexus. *Channels*, **8**(2), 131–141.

Mateos-Aparicio, P., & Rodríguez-Moreno, A. (2019). The impact of studying brain plasticity. *Frontiers in Cellular Neuroscience*, **13**, 66.

Monaghan, D. T., Yao, D., Olverman, H. J., Watkins, J. C., & Cotman, C. W. (1984). Autoradiography of D-2-[3H]amino-5-phosphonopentanoate binding sites in rat brain. *Neuroscience Letters*, **52**, 253–258.

Mulkey, R. M., & Malenka, R. C. (1992). Mechanisms underlying induction of homosynaptic long-term depression in area CA1 of the hippocampus. *Neuron*, **9**(5), 967–975.

Nabavi, S., Kessels, H. W., Alfonso, S., Aow, J., Fox, R., & Malinow, R. (2013). Metabotropic NMDA receptor function is required for NMDA receptor-dependent long-term depression. *Proceedings of the National Academy of Sciences of the United States of America*, **110**(10), 4027–4032.

Pastalkova, E., Serrano, P., Pinkhasova, D., Wallace, E., Fenton, A. A., & Sacktor, T. C. (2006). Storage of spatial information by the maintenance mechanism of LTP. *Science (New York)*, **313**(5790), 1141–1144.

Pougnet, J.-T., Compans, B., Martinez, A., Choquet, D., Hosy, E., & Boué-Grabot, E. (2016). P2X-mediated AMPA receptor internalization and synaptic depression is controlled by two CaMKII phosphorylation sites on GluA1 in hippocampal neurons. *Scientific Reports*, **6**(1), 1.

Qu, Y., Misaghi, S., Newton, K., Gilmour, L. L., Louie, S., Cupp, J. E., Dubyak, G. R., Hackos, D., & Dixit, V. M. (2011). Pannexin-1 is required for ATP release during apoptosis but not for inflammasome activation. *The Journal of Immunology*, **186**(11), 6553–6561.

Romanov, R. A., Bystrova, M. F., Rogachevskaya, O. A., Sadovnikov, V. B., Shestopalov, V. I., & Kolesnikov, S. S. (2012). The ATP permeability of pannexin 1 channels in a heterologous system and in mammalian taste cells is dispensable. *Journal of Cell Science*, **125**(Pt 22), 5514–5523.

Santini, E., Huynh, T. N., & Klann, E. (2014). Mechanisms of translation control underlying long-lasting synaptic plasticity and the consolidation of long-term memory. *Progress in Molecular Biology and Translational Science*, **122**, 131–167.

Stein, I. S., Gray, J. A., & Zito, K. (2015). Non-ionotropic NMDA receptor signaling drives activity-induced dendritic spine shrinkage. *Journal of Neuroscience*, **35**(35), 12303–12308.

Stein, I. S., Park, D. K., Flores, J. C., Jahncke, J. N., & Zito, K. (2020). Molecular mechanisms of non-ionotropic NMDA receptor signaling. *Journal of Neuroscience*, **40**(19), 3741–3750.

Tamburri, A., Dudilot, A., Licea, S., Bourgeois, C., & Boehm, J. (2013). NMDA-receptor activation but not ion flux is required for amyloid-beta induced synaptic depression. *PLoS ONE*, **8**(6), e65350.

Tang, Y. P., Shimizu, E., Dube, G. R., Rampon, C., Kerchner, G. A., Zhuo, M., Liu, G., & Tsien, J. Z. (1999). Genetic enhancement of learning and memory in mice. *Nature*, **401**(6748), 63–69.

Thompson, R. J., Jackson, M. F., Olah, M. E., Rungta, R. L., Hines, D. J., Beazely, M. A., MacDonald, J. F., & MacVicar, B. A. (2008). Activation of pannexin-1 hemichannels augments aberrant bursting in the hippocampus. *Science*, **322**(5907), 1555–1559.

Wang, J., Ambrosi, C., Qiu, F., Jackson, D. G., Sosinsky, G., & Dahl, G. (2014). The membrane protein Pannexin1 forms two open-channel conformations depending on the mode of activation. *Science Signaling*, **7**(335), ra69.

Wang, J., Jackson, D. G., & Dahl, G. (2018). Cationic control of Panx1 channel function. *American Journal of Physiology-Cell Physiology*, **315**(3), C279–C289.

Weilinger, N. L., Lohman, A. W., Rakai, B. D., Ma, E. M. M., Bialecki, J., Maslieieva, V., Rilea, T., Bandet, M. V., Ikuta, N. T., Scott, L., Colicos, M. A., Teskey, G. C., Winship, I. R., & Thompson, R. J. (2016). Metabotropic NMDA receptor signaling couples Src family kinases to pannexin-1 during excitotoxicity. *Nature Neuroscience*, **19**(3), 432–442.

Weilinger, N. L., Tang, P. L., & Thompson, R. J. (2012). Anoxia-induced NMDA receptor activation opens pannexin channels via Src family kinases. *The Journal of Neuroscience: The Official Journal of the Society for Neuroscience*, **32**(36), 12579–12588.

Whitlock, J. R., Heynen, A. J., Shuler, M. G., & Bear, M. F. (2006). Learning induces long-term potentiation in the hippocampus. *Science (New York)*, **313**(5790), 1093–1097.

Yamazaki, Y., Kaneko, K., Fujii, S., Kato, H., & Ito, K.-I. (2003). Long-term potentiation and long-term depression induced by local application of ATP to hippocampal CA1 neurons of the guinea pig. *Hippocampus*, **13**(1), 81–92.

Zhang, J.-M., Wang, H., Ye, C., Ge, W., Chen, Y., Jiang, Z., Wu, C., Poo, M., & Duan, S. (2003). ATP released by astrocytes mediates glutamatergic activity = dependent hetersynaptic suppression. *Neuron*, **40**(5), 971–982.

Zhang, W., Lohman, A. W., Zhuravlova, Y., Lu, X., Wiens, M. D., Hoi, H., Yaganoglu, S., Mohr, M. A., Kitova, E. N., Klassen, J. S., Pantazis, P., Thompson, R. J., & Campbell, R. E. (2017). Optogenetic control with a photocleavable protein, PhoCl. *Nature Methods*, **14**(4), 391–394.

## Additional information

### Competing interests

The authors declare that they have no competing interests.

### Author contributions

A.C.N. and R.J.T. designed the study. A.C.N. performed and analysed all electrophysiology experiments, prepared figures and wrote the manuscript. C.L.A. performed and analysed all ion replacement experiments. A.K.J.B. performed PhoCl and western blot experiments. A.C.N., C.L.A., A.K.J.B. and R.J.T. revised and approved the final manuscript submitted for publication. R.J.T. supervised the study.

### Funding

This study was funded by operating grants from RJT from Canadian Institute for Health Research.

### Keywords

ATP, long-term depression, metabotropic signalling, NMDAR, pannexin-1, purinergic P2X4 receptor

### Supporting information

Additional supporting information can be found online in the Supporting Information section at the end of the HTML view of the article. Supporting information files available:

**Peer Review History**

