## [Peer Review History · The Journal of Physiology]

Non-ionotropic NMDAR signaling activates Panx1 to induce P2X4R-dependent long-term depression in the hippocampus

Allison C Nielsen, Connor L. Anderson, Carina Ens, Andrew K.J. Boyce, and Roger J. Thompson

DOI: 10.1113/JP285193

Corresponding author(s): Roger Thompson (rj.thompson@ucalgary.ca)

Review Timeline:

Submission Date:	01-Apr-2024
Editorial Decision:	29-Apr-2024
Revision Received:	24-Oct-2024
Editorial Decision:	18-Nov-2024
Revision Received:	22-Nov-2024
Editorial Decision:	25-Nov-2024
Revision Received:	26-Nov-2024
Accepted:	28-Nov-2024

Senior Editor: Peking Fong

Reviewing Editor: Peking Fong

Transaction Report:

Dear Dr Nielsen,

Re: JP-RP-2024-285193 "Metabotropic NMDAR activation of Panx1 causes a P2X4-dependent long-term depression in the hippocampus" by Roger J. Thompson, Allison C Nielsen, and Connor L. Anderson

Thank you for submitting your manuscript to The Journal of Physiology. It has been assessed by a Reviewing Editor and by 2 expert referees and we are pleased to tell you that it is potentially acceptable for publication following satisfactory major revision.

Please address all the points raised and incorporate all requested revisions or explain in your Response to Referees why a change has not been made. We hope you will find the comments helpful and that you will be able to return your revised manuscript within 3 months. If you require longer than this, please contact journal staff: jp@physoc.org. Please note that this letter does not constitute a guarantee for acceptance of your revised manuscript.

LANGUAGE EDITING AND SUPPORT FOR PUBLICATION: If you would like help with English language editing, or other article preparation support, Wiley Editing Services offers expert help, including English Language Editing, as well as translation, manuscript formatting, and figure formatting at www.wileyauthors.com/eoo/preparation. You can also find resources for Preparing Your Article for general guidance about writing and preparing your manuscript at www.wileyauthors.com/eoo/prepresources.

REVISION CHECKLIST:

We look forward to receiving your revised submission.

Yours sincerely,

Peying Fong
Senior Editor
The Journal of Physiology

REQUIRED ITEMS

- Author photo and profile. First or joint first authors are asked to provide a short biography (no more than 100 words for one author or 150 words in total for joint first authors) and a portrait photograph. These should be uploaded and clearly labelled together in a Word document with the revised version of the manuscript. See Information for Authors for further details.
- Your manuscript must include a complete Additional Information section, including competing interests; funding; author contributions and acknowledgements.
- Please upload separate high-quality figure files via the submission form.
- Please ensure that the Article File you upload is a Word file.
- Papers must comply with the Statistics Policy: https://jp.msubmit.net/cgi-bin/main.plex?form_type=display_requirements#statistics.

In summary:

- If $n \leq 30$, all data points must be plotted in the figure in a way that reveals their range and distribution. A bar graph with data points overlaid, a box and whisker plot or a violin plot (preferably with data points included) are acceptable formats.
- If $n > 30$, then the entire raw dataset must be made available either as supporting information, or hosted on a not-for-profit repository, e.g. FigShare, with access details provided in the manuscript.
- 'n' clearly defined (e.g. x cells from y slices in z animals) in the Methods. Authors should be mindful of pseudoreplication.
- All relevant 'n' values must be clearly stated in the main text, figures and tables.
- The most appropriate summary statistic (e.g. mean or median and standard deviation) must be used. Standard Error of the Mean (SEM) alone is not permitted.
- Exact p values must be stated. Authors must not use 'greater than' or 'less than'. Exact p values must be stated to three significant figures even when 'no statistical significance' is claimed.
- A Data Availability Statement is required for all papers reporting original data. This must be in the Additional Information section of the manuscript itself. It must have the paragraph heading 'Data Availability Statement'. All data supporting the

results in the paper must be either: in the paper itself; uploaded as Supporting Information for Online Publication; or archived in an appropriate public repository. The statement needs to describe the availability or the absence of shared data. Authors must include in their statement: a link to the repository they have used, or a statement that it is available as Supporting Information; reference the data in the appropriate section(s) of their manuscript; and cite the data they have shared in the References section. Whenever possible, the scripts and other artefacts used to generate the analyses presented in the paper should also be publicly archived. If sharing data compromises ethical standards or legal requirements then authors are not expected to share it, but must note this in their statement. For more information, see our Statistics Policy.

- Please include an Abstract Figure file, as well as the Figure Legend text within the main article file. The Abstract Figure is a piece of artwork designed to give readers an immediate understanding of the research and should summarise the main conclusions. If possible, the image should be easily 'readable' from left to right or top to bottom. It should show the physiological relevance of the manuscript so readers can assess the importance and content of its findings. Abstract Figures should not merely recapitulate other figures in the manuscript. Please try to keep the diagram as simple as possible and without superfluous information that may distract from the main conclusion(s). Abstract Figures must be provided by authors no later than the revised manuscript stage and should be uploaded as a separate file during online submission labelled as File Type 'Abstract Figure'. Please also ensure that you include the figure legend in the main article file. All Abstract Figures should be created using BioRender. Authors should use The Journal's premium BioRender account to export high-resolution images. Details on how to use and access the premium account are included as part of this email.

EDITOR COMMENTS

Reviewing Editor:

I agree with the reviewers that this work holds significant importance for the fields of pannexin and neuroscience. Both reviewers have suggested that further analysis and specific experiments are necessary to consolidate the notion that ATP alone is sufficient to promote niNMDAR/Panx1-dependent LTD. All of these request are addressable. The incorporation of the MK801 blocker in a few experiments would be beneficial. Additionally, both reviewers indicate that Panx1 might still permit calcium ions to permeate under NMDAR-Src activation in the large-pore channel form. While the authors might argue that there is evidence suggesting ATP release can still occur in Panx1 channels with small conductance that are only permeable to chloride (PMID: 33410749), it would still be beneficial to the field if the authors could experimentally address this issue as well.

Please also see 'Required Items' above.

Senior Editor:

The review of your manuscript is now complete, and critiques of two expert Referees and a Reviewing Editor are attached herewith. They agree that your manuscript provides evidence supportive of a schema whereby long-term hippocampal depression involves interplay between non-ionotropic NMDA receptor activation, pannexin-mediated ATP release, and ionotropic purinergic receptor (P2X4) activation. You will read, however, that both Referees also offer detailed suggestions for improvement, despite the study's overall promise. Although many queries can be addressed by re-analyses of data at hand, the Referees and Reviewing Editor feel there is need to incorporate additional key experiments, specified within their comments. The Reviewing Editor expects that these are addressable, and I agree with this assessment. In prioritizing additional experiments, I advise you to direct your focus on those summarized by the Reviewing Editor.

In addition, as you prepare your revised manuscript, please note the Statistical Policy requires use of the standard deviation when presenting mean data, rather than the standard error of the mean used in the previous version.

Thank you for entrusting your manuscript to The Journal of Physiology. I look forward to receiving your revision soon.

REFEREE COMMENTS

Referee #1:

Though past work has linked non-ionotropic NMDAR (niNMDAR) signaling to long-term depression (LTD), the mechanisms that contribute to establishing this form LTD are not completely understood. The manuscript by Nielsen et al. reports on a novel role of Panx1-initiated ATP release in the context of non-ionotropic NMDA receptor (niNMDARs) dependent long-term depression (LTD). The strength of the study comes from the experimental approach, including novel reagents (e.g. TAT-

Panx(Y308)), used to support niNMDAR-induced Panx1 activation. This is based on their past studies, which defined signaling mechanisms through which niNMDARs initiate Panx1 opening. Their findings are potentially important in providing new insights regarding mechanisms that regulate LTD, an important form of NMDAR-dependent synaptic plasticity. However, there are several concerns that should be addressed to further improve the manuscript.

- 1) A number of inhibitors, applied by bath or patch pipette, are used to establish the role of Panx1, Src, etc. The authors should report whether any of these treatments had an effect on the amplitude of EPSCs when compared to control recordings (e.g. could be reported in the text or in table format).
- 2) Responders/non-responders: when reporting the effect of various treatments on induction of LTD, the authors report the outcome for a subset of recordings (e.g. 10/14 in Fig 1a), based on criteria defined in the methods (data analysis and statistics). But in some cases, the sample size is small and the responder rate low (e.g. Fig 3A: 7/11; 4A: 4/6). Some additional analysis and discussion should be included to address this. For example, it may be more suitable to use a non-parametric test to compare these and other data sets. Also, outcomes for non-responders should be discussed. For example, in Fig 3A, 4/11 recordings presumably exhibited LTD in the presence of 10panx, suggesting some heterogeneity in Panx1 dependence for LTD induction (an interesting observation that warrants discussion).
- 3) Fig 5A: in the presence of TAT-Panx(y308) LTP is now evident in response to LFS. This should be mentioned as it suggests that concurrent activation of Panx1 opposes LTP induction, at least in a subset of recordings. Note: on p9, 2nd paragraph, the authors state that EPSCs in the presence of TAT-Panx(y308) were smaller compared to control, when in fact they are larger. This should be corrected.
- 4) Panx1 Ca²⁺ permeability: The authors report that Panx1 is not Ca²⁺ permeable under the reported recording conditions. But Panx1 permeability varies depending on activation state (small (Cl⁻ permeable) vs large (Ca²⁺ and ATP permeable) pore state). More relevant would be to examine Ca²⁺ permeability under conditions that reflect large pore opening induced by niNMDARs (e.g. NMDA-evoked in HEK co-expressing NMDARs). Last, results from recordings of Panx1 in HEK cells, including representative traces, should be shown.
- 5) ATP rescue: Panx1 is well known to contribute to purinergic signaling and these findings suggest an important contribution of Panx1 dependent ATP release to niNMDAR-induced LTD, which is interesting. But ATP may be recruiting a distinct mechanism for LTD. Did ATP alone, without LFS, induced any depression? Or does ATP rescue require concurrent NMDARs stimulation? This could be tested by examining the effect of co-applied APV.

Minor

- 1) Abstract: "In addition to a role in long-term depression (LTD), non-ionotropic NMDARs
- 2) (niNMDAR)." This sentence is incomplete and should be revised.
- 3) Last paragraph of the intro: "In addition to this work on synaptic plasticity," could be made clearer. Suggest the following "In addition to their role in inducing LTD, our work has identified..."
- 4) p4 and p8: References by Goh & Manahan-Vaughan, 2013; Kemp & Manahan-Vaughan, 2004; Manahan-
- 5) Vaughan & Braunewell, 1999 - these studies did not examine the role of Panx1 in LTD and should be deleted.
- 6) p8, Fig 3A: The text reports 9 out 11, but the figure lists 7 out of 11
- 7) p10, 2nd paragraph: "Fig 7D" should be "7C".
- 8) "signaling" or "signalling" should be used consistently throughout.

Referee #2:

The present manuscript, "Metabotropic NMDAR activation of Panx1 causes a P2X4-dependent long-term depression in the hippocampus," by Nielsen et al., investigates the contribution of Panx1 channel activity to excitatory long-term depression. Using electrophysiological recordings, the Authors report evidence for differential involvement of ionotropic and non-ionotropic NMDAR function in LTD and, in this context, provide support for the non-ionotropic NMDARs activity recruiting Src

kinase, Panx1, and purinergic signaling required for sustaining LTD.

A couple of minor concerns and interpretations should be addressed, which would strengthen the discussion of the results and visualize the potential mechanism.

Specific comments

-The Authors demonstrate that 3Hz-LFS stimulation induces reliable LTD in rat hippocampal slices, which is completely prevented by the presence of APV but partially blocked by MK-801 incubation. MK-801 only blocked the initial phase but failed to block the maintenance of the LTD. However, unlike APV, MK801 was only applied during baseline and low-frequency stimulations, being rinsed after that. Authors argued that MK801 displays a slow time constant for their effect recovery, so it remains during the recording. Nevertheless, since niNMDAR-Src-dependent Panx1 activation promotes ATP release and calcium influx via P2X4, it can depolarize membrane potential and eventually promote ionotropic NMDAR activation. Thus, experiments to block the ionotropic function of NMDAR should be done in the presence of MK801 in the entire recording.

-AMPA receptors essentially mediate synaptic responses at basal frequency stimulation. Thus, persistent changes in synaptic strength are supported by changes in the number and function of AMPARs. In this regard, how are niNMDAR-Src-Panx1 and P2X4 activity related to the depression of the evoked synaptic currents?

-Regarding calcium permeation through Panx1, the Authors cannot discard, at least with the present experiments, that Panx1 could mediate Ca influx under niNMDAR-Src activation. First, experiments were done in HEK293, which lacks niNMDAR-Src signaling and Y308 phosphorylation. In this sense, the question arises whether Y308 phosphorylation can induce a large pore conformation of Panx1 channels and calcium entry? Experiments in conditions where NMDAR-Panx1-Src is functional, for instance, in primary or neuron cell line cultures, should be done to probe whether Ca influx is possible through phosphorylated Y308 Panx1. The present results in HEK293 should be included in a figure.

-Authors indicate that exogenous ATP can restore LTD induced by LFS in the presence of the TAT-PanxY308 peptide. In the physiological context, what could be the source of ATP? Is it coming from postsynaptic neurons or glial cells?

-Despite 5BDBD is a selective P2X4 antagonist, experiments supporting P2X4 involvement in NMDAR-dependent LTD require additional experiments such as knockdown or knockout conditions. According to the data in Figure 8, LTD was induced in the presence of 5BDBD, but it was significantly blocked only after 20 minutes of the LFS application. So, does the Panx1 channel activation or the ATP released by Panx1 channels occur after this time? Or is this the time it takes to reach ATP levels to activate P2X4?

P2X receptors exhibit different ATP affinities; therefore, can 3Hz LFS-LTD require the activation of another P2X receptor, such as P2X7? In this sense, how to explain the potentiation observed in 10panx treated slices?

Does the calcium entry by P2X4 contribute to the activation of Ca-dependent phosphatases and AMPAR removal to support the depression of EPSC?

-The inclusion of a model of the present data would be illustrative for readers

Minor points:

-In the abstract section, page 2, line 2: the second phrase seems incomplete.

-On page 2, line 3: change "contribute" by "contributes".

-On page 4, line 12: "Due to recent findings linking Panx1 to LTD..." the following references nor address the link between Panx1 and LTD: Goh & Manahan-Vaughan, 2013; Kemp & Manahan-Vaughan, 2004; Manahan-Vaughan & Braunewell,

1999.

-On page 10, lines 6-7: "Application of ATP alone can induce synaptic depression... reference Vanden Abeele et al., 2006; this article does not address the involvement of ATP in LTD.

-On page 9, line 9:" EPSCs in the presence of TAT-PanxY308 were

significantly smaller compared to control without peptide", it should be significantly greater compared to... since EPSC amplitude was higher after LFS in the TAT-PanxY308 group.

-On page 8, lines 1-2: "In the first 5 min after LFS, MK801 prevented..." Does this comparison refer to Figure 2A? Please indicate.

-On page 8, line 5:" This depression was statistically different than D-APV at 20 min." Does this comparison refer to Figure 1E? Please indicate.

END OF COMMENTS

Referee #1:

Though past work has linked non-ionotropic NMDAR (niNMDAR) signaling to long-term depression (LTD), the mechanisms that contribute to establishing this form LTD are not completely understood. The manuscript by Nielsen et al. reports on a novel role of Panx1-initiated ATP release in the context of non-ionotropic NMDA receptor (niNMDARs) dependent long-term depression (LTD). The strength of the study comes from the experimental approach, including novel reagents (e.g. TAT-Panx(Y308)), used to support niNMDAR-induced Panx1 activation. This is based on their past studies, which defined signaling mechanisms through which niNMDARs initiate Panx1 opening. Their findings are potentially important in providing new insights regarding mechanisms that regulate LTD, an important form of NMDAR-dependent synaptic plasticity. However, there are several concerns that should be addressed to further improve the manuscript.

1) A number of inhibitors, applied by bath or patch pipette, are used to establish the role of Panx1, Src, etc. The authors should report whether any of these treatments had an effect on the amplitude of EPSCs when compared to control recordings (e.g. could be reported in the text or in table format).

We have performed a non-parametric (Wilcoxon) test on the effects of the blockers on EPSC amplitude, comparing 5 min of drug treatment to the control and have included these results in the appropriate sections. We did not detect significant effects on the EPSC amplitude by the drugs themselves.

2) Responders/non-responders: when reporting the effect of various treatments on induction of LTD, the authors report the outcome for a subset of recordings (e.g. 10/14 in Fig 1a), based on criteria defined in the methods (data analysis and statistics). But in some cases, the sample size is small and the responder rate low (e.g. Fig 3A: 7/11; 4A: 4/6). Some additional analysis and discussion should be included to address this. For example, it may be more suitable to use a non-parametric test to compare these and other data sets. Also, outcomes for non-responders should be discussed. For example, in Fig 3A, 4/11 recordings presumably exhibited LTD in the presence of 10panx, suggesting some heterogeneity in Panx1 dependence for LTD induction (an interesting observation that warrants discussion).

We have changed the data analysis to use paired non-parametric tests (i.e. Wilcoxon test) as suggested by the reviewer and updated the text. No significant alterations were found compared to our initial use of parametric statistics. We have also added a brief discussion about the outcomes of the non-responders, with appropriate reference that has suggested the past history of the cell changes whether Panx1 contributes to LTP or LTD.

3) Fig 5A: in the presence of TAT-Panx(y308) LTP is now evident in response to LFS. This should be mentioned as it suggests that concurrent activation of Panx1 opposes LTP induction, at least in a subset of recordings. Note: on p9, 2nd paragraph, the authors state that EPSCs in the

presence of TAT-Panx(y308) were smaller compared to control, when in fact they are larger. This should be corrected.

We thank the reviewer for this comment. When we change the statistical analysis to non-parametric, as suggested by the reviewer, the change in EPSC amplitude by TAT-Panx(308) is no longer statistically significant. The text and figure have been changed accordingly.

4) Panx1 Ca²⁺ permeability: The authors report that Panx1 is not Ca²⁺ permeable under the reported recording conditions. But Panx1 permeability varies depending on activation state (small (Cl⁻ permeable) vs large (Ca²⁺ and ATP permeable) pore state). More relevant would be to examine Ca²⁺ permeability under conditions that reflect large pore opening induced by niNMDARs (e.g. NMDA-evoked in HEK co-expressing NMDARs). Last, results from recordings of Panx1 in HEK cells, including representative traces, should be shown.

We thank the reviewer for this comment. Our previous work has shown that Src phosphorylation of Panx1 induces cell death, presumably by activating the large conductance state of the channel. To test this, we developed a light activatable Src kinase, based on our previously published photocleavable (PhoCl) strategy. When Panx1 is expressed with PhoCl-Src in HEK cells, we can increase currents but still do not see a shift in E_{rev} in 40 mM Ca²⁺. This suggests that Src phosphorylation alone is not likely to alter Panx1 selectivity. These data, and the original selectivity data have been combined into a new Figure 6.

5) ATP rescue: Panx1 is well known to contribute to purinergic signaling and these findings suggest an important contribution of Panx1 dependent ATP release to niNMDAR-induced LTD, which is interesting. But ATP may be recruiting a distinct mechanism for LTD. Did ATP alone, without LFS, induced any depression? Or does ATP rescue require concurrent NMDARs stimulation? This could be tested by examining the effect of co-applied APV.

We now show in revised Figure 7 that ATP with D-APV and no 3 Hz synaptic stimulation is sufficient to cause depression. This indicates that ATP alone is sufficient to cause LTD.

Minor

1) Abstract: "In addition to a role in long-term depression (LTD), non-ionotropic NMDARs (niNMDAR)." This sentence is incomplete and should be revised.

Thank you. The sentence has been revised.

3) Last paragraph of the intro: "In addition to this work on synaptic plasticity," could be made clearer. Suggest the following "In addition to their role in inducing LTD, our work has identified..."

Thank you. The sentence has been revised.

4) p4 and p8: References by Goh & Manahan-Vaughan, 2013; Kemp & Manahan-Vaughan, 2004; Manahan- Vaughan & Braunewell, 1999 - these studies did not examine the role of Panx1 in LTD and should be deleted.

The references have been deleted.

6) p8, Fig 3A: The text reports 9 out of 11, but the figure lists 7 out of 11
The text has been revised to 7 of 11 cells.

7) p10, 2nd paragraph: "Fig 7D" should be "7C".
Figure 7 has been revised and the reference in the text is now correct.

8) "signaling" or "signalling" should be used consistently throughout.
Signaling has been replaced with signalling

Referee #2:

The present manuscript, "Metabotropic NMDAR activation of Panx1 causes a P2X4-dependent long-term depression in the hippocampus," by Nielsen et al., investigates the contribution of Panx1 channel activity to excitatory long-term depression. Using electrophysiological recordings, the Authors report evidence for differential involvement of ionotropic and non-ionotropic NMDAR function in LTD and, in this context, provide support for the non-ionotropic NMDARs activity recruiting Src kinase, Panx1, and purinergic signaling required for sustaining LTD.

A couple of minor concerns and interpretations should be addressed, which would strengthen the discussion of the results and visualize the potential mechanism.

Specific comments

-The Authors demonstrate that 3Hz-LFS stimulation induces reliable LTD in rat hippocampal slices, which is completely prevented by the presence of APV but partially blocked by MK-801 incubation. MK-801 only blocked the initial phase but failed to block the maintenance of the LTD. However, unlike APV, MK801 was only applied during baseline and low-frequency stimulations, being rinsed after that. Authors argued that MK801 displays a slow time constant for their effect recovery, so it remains during the recording. Nevertheless, since niNMDAR-Src-dependent Panx1 activation promotes ATP release and calcium influx via P2X4, it can depolarize membrane potential and eventually promote ionotropic NMDAR activation. Thus, experiments to block the ionotropic function of NMDAR should be done in the presence of MK801 in the entire recording.

We apologize for the confusion in our presentation of the data. Both APV and MK-801 were applied transiently, 10 min before and during the LFS stimulation. Followed by wash out. We have altered the text and figures to make the timing of application clearer. At the reviewer's suggestion, we performed experiments where MK-801 was present throughout the recording, which are now included in Fig. 1. In this scenario, LTD was completely blocked. This is an

interesting finding, since transient APV prevents LTD but transient MK-801 does not. We agree with the reviewer that this is likely occurring because P2X4 is activated by ATP released from Panx1, which could either directly cause LTD or indirectly through subsequent activation of ionotropic NMDARs. We have included these possibilities in the Discussion.

-AMPA receptors essentially mediate synaptic responses at basal frequency stimulation. Thus, persistent changes in synaptic strength are supported by changes in the number and function of AMPARs. In this regard, how are niNMDAR-Src-Panx1 and P2X4 activity related to the depression of the evoked synaptic currents?

The exact mechanism is unknown. It is likely that it involves phosphorylation of AMPARs and their subsequent endocytosis. We have added this speculation to the Discussion.

-Regarding calcium permeation through Panx1, the Authors cannot discard, at least with the present experiments, that Panx1 could mediate Ca influx under niNMDAR-Src activation. First, experiments were done in HEK293, which lacks niNMDAR-Src signaling and Y308 phosphorylation. In this sense, the question arises whether Y308 phosphorylation can induce a large pore conformation of Panx1 channels and calcium entry? Experiments in conditions where NMDAR-Panx1-Src is functional, for instance, in primary or neuron cell line cultures, should be done to probe whether Ca influx is possible through phosphorylated Y308 Panx1. The present results in HEK293 should be included in a figure.

We thank the reviewer for this comment, which was also raised by Reviewer 1. Our previous work has shown that Src phosphorylation of Panx1 induces cell death, presumably by activating the large conductance state of the channel. To test this, we developed a light activatable Src kinase, based on our previously published photocleavable (PhoCl) strategy. When Panx1 is expressed with PhoCl-Src in HEK cells, we can increase currents but still do not see a shift in Erev in 40 mM Ca²⁺. This suggests that Src phosphorylation alone is not likely to alter Panx1 selectivity. These data, and the original selectivity data have been combined into a new Fig X.

-Authors indicate that exogenous ATP can restore LTD induced by LFS in the presence of the TAT-PanxY308 peptide. In the physiological context, what could be the source of ATP? Is it coming from postsynaptic neurons or glial cells?

We do not know the source of the ATP, but the most likely explanation is release through Panx1 in the postsynaptic neuron. It is possible that Panx1 may release other factors that stimulate ATP release from nearby astrocytes. This possibility has been noted in the discussion.

-Despite 5BDBD is a selective P2X4 antagonist, experiments supporting P2X4 involvement in NMDAR-dependent LTD require additional experiments such as knockdown or knockout conditions. According to the data in Figure 8, LTD was induced in the presence of 5BDBD, but it was significantly blocked only after 20 minutes of the LFS application. So, does the Panx1 channel activation or the ATP released by Panx1 channels occur after this time? Or is this the time it takes to reach ATP levels to activate P2X4?

We thank the reviewer for the suggestion of knockdown / out experiments. This was

unfortunately not feasible because of the length of time to obtain these animals at sufficient numbers. We believe that ATP release is occurring later in the recording after Panx1 is activated by Src. In ischemic conditions, Src activation of Panx1 took several minutes to occur. We do not yet understand why this occurs but have noted it in the text as rationale for the experiment.

P2X receptors exhibit different ATP affinities; therefore, can 3Hz LFS-LTD require the activation of another P2X receptor, such as P2X7? In this sense, how to explain the potentiation observed in 10panx treated slices?

This is an interesting notion that we did not directly pursue because of the occlusion of the response by 5-BDBD, which indicates that a single P2X receptor is sufficient.

Does the calcium entry by P2X4 contribute to the activation of Ca-dependent phosphatases and AMPAR remotion to support the depression of EPSC?

We do not know if this is the case and is the focus of ongoing work.

-The inclusion of a model of the present data would be illustrative for readers

A model has been included.

Minor points:

-In the abstract section, page 2, line 2: the second phrase seems incomplete.

This has been revised accordingly.

-On page 2, line 3: change "contribute" by "contributes".

The change has been made.

-On page 4, line 12: "Due to recent findings linking Panx1 to LTD..." the following references nor address the link between Panx1 and LTD: Goh & Manahan-Vaughan, 2013; Kemp & Manahan-Vaughan, 2004; Manahan-Vaughan & Braunewell, 1999.

Thank you. The references have been deleted.

-On page 10, lines 6-7: "Application of ATP alone can induce synaptic depression... reference Vanden Abeele et al., 2006; this article does not address the involvement of ATP in LTD.

The reference has been removed

-On page 9, line 9: " EPSCs in the presence of TAT-PanxY308 were significantly smaller compared to control without peptide", it should be significantly greater compared to... since EPSC amplitude was higher after LFS in the TAT-PanxY308 group.

This has been corrected.

-On page 8, lines 1-2: "In the first 5 min after LFS, MK801 prevented..." Does this comparison refer to Figure 2A? Please indicate.

Correct. This has been clarified.

-On page 8, line 5:" This depression was statistically different than D-APV at 20 min." Does this comparison refer to Figure 1E? Please indicate.

Fig 1 has been revised with new data and comparisons in the text should now be clear

Dear Dr Thompson,

Re: JP-RP-2024-285193R1 "Metabotropic NMDAR activation of Panx1 causes a P2X4-dependent long-term depression in the hippocampus" by Allison C Nielsen, Connor L. Anderson, Carina Ens, Andrew KJ Boyce, and Roger J. Thompson

Thank you for submitting your manuscript to The Journal of Physiology. It has been assessed by a Reviewing Editor and by 2 expert referees and we are pleased to tell you that it is acceptable for publication following satisfactory minor revision.

REVISION CHECKLIST:

We look forward to receiving your revised submission.

Yours sincerely,

Peying Fong
Senior Editor
The Journal of Physiology

EDITOR COMMENTS

Reviewing Editor:

The authors have addressed the primary concerns raised by the reviewers. The minor recommendations can be readily addressed by the authors.

Senior Editor:

The review of your manuscript is now complete.

Both Expert Referees and the Reviewing Editor agree that concerns raised in previous review are satisfactorily addressed in this version, and moreover, that the manuscript is significantly improved.

There are, however, several minor points raised by the referees. Both the Reviewing Editor and I agree these can be readily addressed by your team.

In addition, although data presented indeed comply with the statistics policy of The Journal of Physiology (that is, as mean \pm SD), the section within the Methods pertaining to "Data analysis and statistics" (page 7) states otherwise: "...results represented as mean \pm SEM." As you prepare your revision, please also make sure to correct this oversight.

We look forward to receiving your revised manuscript. Thank you for contributing your work to The Journal of Physiology.

REFeree COMMENTS

Referee #1:

The revisions, including additional analyses and data, and response letter address my major concerns. I only have the following suggestions:

- 1) P10: for results presented in Fig 5 and 6, the authors conclude that results have bearing for mechanisms underlying "LTD after transient MK801". However, MK801 was not transiently applied in these experiments. This should be restated to avoid confusion.
- 2) P15, last sentence: "...NMDARs because D-APV prevented the depression of EPSCs by ATP alone." Should be corrected. Believe they meant to say "D-APV did not prevent" as shown in Fig 7

Referee #2:

The present manuscript by Nielsen et al., investigates the contribution of Panx1 channel activity to hippocampal excitatory long-term depression (LTD). Using electrophysiological recordings and pharmacological and molecular approaches, the authors report a new LTD mechanism in hippocampal slices that involves non-ionotropic NMDARs activity, the phosphorylation of Panx1 at Y308 residue by Src kinase, the consequent ATP release by Panx1 and P2X4 activation. The article is original, well-written, and proposes an intriguing new mechanism. In the revised manuscript version, the authors have tried to address most of my questions. The corrections and edits included in the present version improved both in terms of the presentation of the data and by the inclusion of a few novel experiments. The arguments exposed by the authors are, in my opinion, reasonable, and they alleviated many of my concerns.

Minor points:

In Figures 1A, E, G, and 7C, the representative traces of EPSCs seem ungrouped and unconstructed.

In page 4, line 2: "current (Fig 6F; 227.8 {plus minus} 41.9 % baseline; p = 0.0221 (*) with Dunnett's multiple comparison post-test", it refers to Fig 6G

END OF COMMENTS

We thank the reviewers and editors for their careful consideration of our manuscript. We have made the minor revisions as suggested by the reviewers and updated the representative traces in the figures.

Dear Dr Thompson,

Re: JP-RP-2024-285193R2 "Non-ionotropic NMDAR signaling activates Panx1 to induce P2X4R-dependent long-term depression in the hippocampus" by Allison C Nielsen, Connor L. Anderson, Carina Ens, Andrew K.J. Boyce, and Roger J. Thompson

Thank you for submitting your manuscript to The Journal of Physiology. It has been assessed by a Reviewing Editor and by 0 expert referees and we are pleased to tell you that it is acceptable for publication following satisfactory minor revision.

The review comments are copied at the end of this email.

REVISION CHECKLIST:

We look forward to receiving your revised submission.

Yours sincerely,

Peying Fong
Senior Editor
The Journal of Physiology

EDITOR COMMENTS

Thank you for your responsiveness to the critiques provided in the last review cycle.

An additional section appears to have been incorporated in the Methods, de novo, on page 5 of the manuscript proper ("Plasmids"). This of course is fine, however there seems to be a new reference (appearing in boldface) to a 2005 paper by Bruzzone et al. This, however, does not appear in the References; please ensure it is incorporated. In the following sentence, please also remove the word "here" when referencing Zhang, 2017. In the present context, it is misleading. As it stands, the phrasing gives the impression that the PhoCI construct is described within the present manuscript, and incorrectly suggests the reference is out of place.

END OF COMMENTS

We thank the reviewers and editors for their careful consideration of our manuscript. We have made the minor revisions as suggested by the reviewers and updated the representative traces in the figures.

EDITOR COMMENTS

Thank you for your responsiveness to the critiques provided in the last review cycle.

An additional section appears to have been incorporated in the Methods, de novo, on page 5 of the manuscript proper ("Plasmids"). This of course is fine, however there seems to be a new reference (appearing in boldface) to a 2005 paper by Bruzzone et al. This, however, does not appear in the References; please ensure it is incorporated.

Thank you for pointing out this oversight. We have included the reference and properly formatted the text.

In the following sentence, please also remove the word "here" when referencing Zhang, 2017. In the present context, it is misleading. As it stands, the phrasing gives the impression that the PhoCI construct is described within the present manuscript, and incorrectly suggests the reference is out of place.

The sentence has been rephrased to ensure the source of the PhoCI construct and its original description is clear.

Dear Dr Thompson,

Re: JP-RP-2024-285193R3 "Non-ionotropic NMDAR signaling activates Panx1 to induce P2X4R-dependent long-term depression in the hippocampus" by Allison C Nielsen, Connor L. Anderson, Carina Ens, Andrew K.J. Boyce, and Roger J. Thompson

We are pleased to tell you that your paper has been accepted for publication in The Journal of Physiology.

Yours sincerely,

Peying Fong
Senior Editor
The Journal of Physiology

If you would like to receive our 'Research Roundup', a monthly newsletter highlighting the cutting-edge research published in The Physiological Society's family of journals (The Journal of Physiology, Experimental Physiology, Physiological Reports, The Journal of Nutritional Physiology and The Journal of Precision Medicine: Health and Disease), please click this link, fill in your name and email address and select 'Research Roundup':
<https://www.physoc.org/journals-and-media/membernews>

- You can help your research get the attention it deserves! Check out Wiley's free Promotion Guide for best-practice recommendations for promoting your work at: www.wileyauthors.com/eeo/guide. You can learn more about Wiley Editing Services which offers professional video, design, and writing services to create shareable video abstracts, infographics, conference posters, lay summaries, and research news stories for your research at: www.wileyauthors.com/eeo/promotion.

EDITOR COMMENTS

Thank you for satisfactorily incorporating these last details. I look forward to final publication of your manuscript. Congratulations, and thank you for contributing your work to The Journal of Physiology.